# Anthocyanin Intake and Physical Activity: Associations with the Lipid Profile of a US Working Population

**DOI:** 10.3390/molecules25194398

**Published:** 2020-09-24

**Authors:** Maria S. Hershey, Mercedes Sotos-Prieto, Miguel Ruiz-Canela, Miguel Angel Martinez-Gonzalez, Aedin Cassidy, Steven Moffatt, Stefanos N. Kales

**Affiliations:** 1Department of Preventive Medicine and Public Health, Navarra Institute for Health Research, University of Navarra, 31008 Pamplona, Spain; mhershey@alumni.unav.es (M.S.H.); mcanela@unav.es (M.R.-C.); mamartinez@unav.es (M.A.M.-G.); 2Department of Environmental Health, Harvard T.H. Chan School of Public Health, 677 Huntington Avenue, Boston, MA 02115, USA; msotosp@hsph.harvard.edu; 3Department of Preventive Medicine and Public Health, School of Medicine, Universidad Autónoma de Madrid, 28049 Madrid, Spain; 4CIBER of Epidemiology and Public Health (CIBERESP), Carlos III Health Institute, 28029 Madrid, Spain; 5Biomedical Research Network Centre for Pathophysiology of Obesity and Nutrition (CIBEROBN), Carlos III Health Institute, 28029 Madrid, Spain; 6Department of Nutrition, Harvard T.H. Chan School, Boston, MA 02115, USA; 7The Institute for Global Food Security, Queen’s University Belfast, Northern Ireland BT9 7BL, UK; A.Cassidy@qub.ac.uk; 8National Institute for Public Safety Health, Indianapolis, IN 324 E New York Street, Indianapolis, IN 46204, USA; steven.moffatt@ascension.org; 9Occupational Medicine, The Cambridge Health Alliance/Harvard Medical School, Boston, MA 02139, USA

**Keywords:** anthocyanins, physical activity, lipid profile, cardiovascular disease, working population

## Abstract

While growing evidence exists on the independent associations between anthocyanins and physical activity on cardiovascular disease (CVD) risk determinants, the possible interaction between these exposures has not yet been studied. We aimed to study the potential synergism between anthocyanin intake and physical activity on lipid profile measures. This cross-sectional study was conducted among 249 US career firefighters participating in the *Feeding America’s Bravest* trial. Anthocyanin intake was calculated using a validated food frequency questionnaire (FFQ) and physical activity level by a validated questionnaire. Multivariable linear regression models determined the extent to which anthocyanin intake and physical activity predicted lipid parameters. Generalized linear models were used for joint effect and interaction analyses on the multiplicative and additive scales. Both anthocyanins and physical activity were independently inversely associated with total cholesterol:high density lipoprotein (HDL) cholesterol. Only physical activity was inversely associated with triglycerides, low density lipoprotein (LDL) cholesterol:HDL, and triglycerides (TG):HDL. Although the combined exposure of low anthocyanin intake and low physical activity was associated with lower (RR = 2.83; 95% CI: 1.42 to 5.67) HDL cholesterol <40 mg/dL, neither multiplicative (*p* = 0.72) nor additive interactions were detected (relative excess risk due to interaction (RERI): 0.02; 95% CI: −1.63 to 1.66; *p* = 0.98). Our findings provide insight on the potential synergism between anthocyanin intake and physical activity on the lipid profile.

## 1. Introduction

The prevalence of low fitness, obesity, and cardiovascular disease (CVD) risk factors, such as hypercholesterolemia, hypertension, and high blood glucose, among United States (US) career firefighters is high [1]. When these risk factors interact with the strenuous physical activity, emotional stress, and environmental pollutants characteristic of the firefighter profession, the risk of CVD events is increased [2]. Sudden cardiac death is the leading cause of on-duty deaths, 82% of which are attributed to underlying coronary heart disease and cardiomegaly/left ventricular hypertrophy; moreover, CVD contributes to important morbidity and disability among this working population [2,3,4,5]. Therefore, the primary prevention of CVD through the promotion of healthy dietary patterns and physical activity should be a current priority. Polyphenols in particular play an important role as one of the dietary components associated with the cardioprotective effects of certain foods common across different healthy diets.

Polyphenols are secondary plant metabolites and bioactive compounds naturally occurring in plants and plant-derived products, which can be differentiated into six main classes: flavones, flavonols, flavanols, flavanones, anthocyanins, and flavan-3-ols. Anthocyanins are most abundant in red-, purple-, or blue-pigmented plants, flowers, seeds, fruits, and other plant-based foods. A combination of animal and human studies have attributed cardioprotective effects to anthocyanins, including the inhibition of platelet aggregation, increased HDL, arterial vasorelaxation, and improvement of lipid profile and platelet function [6]. These cardioprotective effects suggest biological pathways in which an anthocyanin-rich diet contributes towards the prevention of CVD. A systematic review of randomized controlled trials (RCTs) suggested anthocyanins may have the potential to influence CVD development and progression among individuals with hyperlipidemia [7]. Among men, a prospective cohort study observed a higher intake of anthocyanins was associated with a 14% lower risk of nonfatal myocardial infarction [8]. Evidence suggests positive potential for the use of anthocyanins in the prevention and treatment of CVD risk factors among firefighters.

Anthocyanins have also been associated with nitric oxide production, exercise performance, and physiological responses before, during, and post-exercise, suggesting their antioxidant, anti-inflammatory, and vasoactive properties improve fitness performance [9,10]. Understanding the intercorrelation and combined effect of dietary bioactive components, such as anthocyanins, with physical activity on CVD risk parameters may contribute towards better prevention and treatment of CVD in high-risk populations. Therefore, we aimed to study the joint effect and the possible synergism between anthocyanin intake and physical activity on the lipid profile of Midwestern US career firefighters.

## 2. Results

### 2.1. Study Participants

Among all 249 participants, 95% were male and the mean (±standard deviation (SD)) age was 47 ± 7.6. The average total energy intake among males was 2395 kcal/day, whereas women consumed an average of 1886 kcal/day. The average body mass index (BMI) among participants was 29.8 kg/m^2^. The mean (±SD) values for lipid profile measures at baseline were as follows: TG = 126.37 ± 68.57 mg/dL, total cholesterol = 195.40 ± 36.24 mg/dL, HDL cholesterol = 48.50 ± 10.83 mg/dL, LDL cholesterol = 121.85 ± 31.66 mg/dL, LDL:HDL ratio = 2.61 ± 0.81, TG:HDL ratio = 2.90 ± 2.15, and total cholesterol:HDL ratio = 4.18 ± 1.02.

Baseline characteristics of the overall participants (n = 249) from *Feeding America’s Bravest* included in this study are presented in Table 1 according to low and high anthocyanin intake and physical activity level. Across both subgroups, high anthocyanin intake and physical activity level observed significantly greater mMDS scores and supplement use than those with low intake or activity. Total flavonoid content, protein intake, whole grains, total fiber, polyunsaturated fat, and alcohol consumption were significantly higher among high anthocyanin intakes. Meanwhile, added sugar intake and prevalent dyslipidemia were significantly lower in the high anthocyanin subgroup. Age, BMI, and hours sitting per week were significantly lower among participants with high physical activity levels.

### 2.2. Characteristics of Anthocyanin Intake

Blueberries were the richest source of anthocyanins contributing 43% of total anthocyanin intake (Table 2). Raisins and grapes represented the second richest source (15%); however, they showed the least between-person variability. Strawberries, red wine, apples, and pears followed in contribution and descending between-person variability, respectively. The composition of anthocyanin subclasses was as follows: 27% malvidin, 24% cyanidin, 18% delphinidin, 17% pelargonidin, 10% petunidin, and 4% peonidin, according to the above-mentioned anthocyanin-rich sources, which accounted for 88% of total anthocyanins.

### 2.3. Individual Associations between Anthocyanin Intake and Physical Activity on Lipid Profile

Figure 1 shows regression coefficients for each independent association of anthocyanin intake and physical activity on lipid concentrations and lipid ratios. Multivariable-adjusted models for anthocyanin intake were inversely associated with total cholesterol:HDL (β = −0.14; 95% CI: −0.27 to −0.01, *p* = 0.04) (Appendix A). Although not statistically significant, inverse associations were observed between anthocyanin intake and TG, total cholesterol, LDL cholesterol, and LDL:HDL and TG:HDL ratios and a positive association was observed with HDL cholesterol. After stratifying by low and high physical activity level in Table 3, the association with HDL cholesterol was stronger among the high physical activity subgroup for the age-, sex-, and energy intake-adjusted model (β = 1.97; 95% CI: 0.25 to 3.69) compared to participants with low physical activity (β = 0.52; 95% CI: −1.51 to 2.55); however, this association was lost in multivariable adjusted models. A clear association for anthocyanin intake on total cholesterol:HDL was initially observed across high physical activity; however, there was no clear difference between low (β = −0.16; 95% CI: −0.37 to 0.06) and high physical activity (β = −0.17; 95% CI: −0.32 to −0.01) in the age-, sex-, and energy intake-adjusted model. Nonetheless, statistical significance was lost after additional adjustments for total energy intake, mMDS, multivitamin use, supplement use, and sleep.

Appendix A shows physical activity was inversely associated with TG (β = −5.54; 95% CI: −10.40 to −0.67, *p* = 0.03), LDL:HDL ratio (β = −0.09; 95% CI: −0.14 to −0.03, *p* = 0.005), TG:HDL ratio (β = −0.18; 95% CI: −0.33 to −0.03, *p* = 0.02), and total cholesterol:HDL ratio (β = −0.10; 95% CI: −0.18 to −0.03, *p* = 0.005) after multivariable adjustments. Although not statistically significant, we observed an inverse association of physical activity with total cholesterol and LDL cholesterol; meanwhile, a non-significant positive association was observed with HDL cholesterol. Physical activity stratified by anthocyanin intake in Table 4 shows physical activity was associated with LDL:HDL among low anthocyanin intake (β = −0.10; 95% CI: −0.17 to −0.03) and high anthocyanin intake (β = −0.11; 95% CI: −0.19 to −0.03) in the least adjusted model. Similarly, the total cholesterol:HDL ratio among high anthocyanin intake (β = −0.14; 95% CI: −0.23 to −0.04) and low anthocyanin intake (β = −0.14; 95% CI: −0.23 to −0.05) observed similar associations when adjusted for age, sex, and energy intake. Overall, the associations observed in the stratified analysis were consistent across subgroups, suggesting the effect of physical activity was independent of anthocyanin intake. Sensitivity analysis for both exposures with additional exclusions for chronic disease, women, and supplement use further supported the robustness of our findings (Appendix A).

### 2.4. The Combined Effect, Stratification, and Interaction Analyses between Anthocyanin Intake and Physical Activity on HDL Cholesterol <40 mg/dL

Table 5 shows the four combined effects created between low and high anthocyanin intake and physical activity on HDL cholesterol <40 mg/dL. The combined effects were as follows: low anthocyanins/high activity (RR = 1.46; 95% CI: 0.68 to 3.11, *p* = 0.33), high anthocyanins/low activity (RR = 2.36; 95% CI: 1.15 to 4.83, *p* = 0.02), and low anthocyanins/low activity (RR = 2.83; 95% CI: 1.42 to 5.67, *p* = 0.003). Consistent with the prevalence observed within each subgroup, low anthocyanin intake in combination with low physical activity was associated with the highest relative risk. Stratification analysis observed significant associations between physical activity among those with high anthocyanin intake (RR = 2.19; 95% CI: 1.07 to 4.49, *p* = 0.03) and low anthocyanin intake (RR = 1.99; 95% CI: 1.10 to 3.60, *p* = 0.02). Although neither interactions on the multiplicative scale (*p* = 0.72) nor the relative risk due to interaction on the additive scale (RERI: 0.02; 95% CI: −1.63 to 1.66; *p* = 0.98) were statistically significant, the joint effect analysis suggested the combination of low anthocyanin intake and low physical activity was associated with the greatest risk compared to the individual effects of the two exposures.

## 3. Discussion

To our knowledge, this is the first study to assess both individual and joint exposures of anthocyanin intake and physical activity on lipid profile measures and test for multiplicative and additive measures of interaction. Independently, anthocyanins and physical activity were both inversely associated with total cholesterol:HDL, whereas only physical activity was inversely associated with triglycerides, LDL:HDL, and TG:HDL. Furthermore, the combined effect of a low anthocyanin intake and low physical activity more than doubled the relative risk of having HDL cholesterol <40 mg/dL, compared to the reference high anthocyanins/high activity joint exposure, although no statistically significant interaction was observed.

Berries, in particular, have been studied for their rich anthocyanin content; meanwhile, other studies have used anthocyanin supplementation or derived total anthocyanin intake from the habitual diet [11,12,13,14,15]. Some evidence suggests that the effect of polyphenols may depend on the form in which they are consumed in such a way that supplements may not offer the same synergistic effects and health benefits as food sources [16]. Our top five sources of anthocyanin-rich foods included blueberries, strawberries, red wine, apple or pears, and raisins or grapes, which accounted for 88% of total anthocyanin intake. The remaining 12% can be attributed to other sources, including peaches or plums, apricots, bananas, cantaloupe, and prunes, which contributed small amounts to the total anthocyanin intake in our study. Additional sources of anthocyanins, which were not collected in the study FFQ, include some red to purplish blue-colored vegetables and grains, such as purple corn, purple sweet potato, red cabbage, black carrot, black soybean, and some varieties of rice [17]. Our analysis on anthocyanin intake suggested the greatest between-person variability of anthocyanin intake was associated with the consumption of blueberries and strawberries, most likely due to the seasonality and affordability aspects of purchasing berries.

A variety of scientific studies, including in vitro studies, animal models, and human clinical trials, show that anthocyanins possess anti-inflammatory and antimicrobial activities, which improve cardiometabolic, visual, and neurological health. These protective effects have been explained by participation in different mechanisms and pathways, including the free-radical scavenging pathway, cyclooxygenase pathway, mitogen-activated protein kinase pathway, and inflammatory cytokines signaling, as well as some crucial cellular processes, including the cell cycle, apoptosis, autophagy, and biochemical metabolism [17,18]. Protective effects of anthocyanin intake previously associated with CVD risk determinants further support our findings. A meta-analysis of RCTs using purified anthocyanins or anthocyanins-rich foods as treatment compared with a placebo or nonexposed controls observed anthocyanins significantly reduced total cholesterol (standardized mean difference (SMD): −0.33; 95% CI: −0.62, −0.03; I2 = 86.9%), and LDL cholesterol (SMD: −0.35; 95% CI: −0.66, −0.05; I2 = 85.2%) in regards to the lipid profile [19]. A more recent meta-analysis of 45 randomized controlled trials stated that the consumption of berries and purified anthocyanins (2.2−1230 mg anthocyanins/day) significantly increased HDL cholesterol and reduced total cholesterol, LDL cholesterol, and TGs [14]. Similar to our findings for anthocyanin intake, which lost statistical significance in multivariable models, a systematic review noted that most of the potential effects observed across RCTs were nonsignificant. However, improvement of biomarkers were consistent across studies, particularly in those with elevated lipids at baseline [7].

Physical activity showed clear inverse associations with triglycerides, LDL:HDL, TG:HDL, and total cholesterol:HDL when measured with an ordinal scale capturing habitual weekly frequency and intensity of physical activity and exercise performed. Existing evidence supports aerobic exercise of adequate intensity, duration, and volume results in favorable and independent improvements of blood lipids and lipoproteins in individuals with and without dyslipidemia. Furthermore, the most consistent findings are shown for increases in HDL cholesterol [20,21,22]. A randomized controlled trial suggested improvements in lipids and lipoproteins were related to the amount of activity and not to the intensity of exercise or improvement in fitness [23]. In particular, a relatively high amount of regular exercise (equivalent to jogging 27.2 to 28.8 km/week at moderate pace) significantly improved the overall lipoprotein profile by decreasing LDL size, increasing HDL cholesterol concentration and size, and decreasing triglycerides, which was not observed for lower amounts of exercise [23]. Therefore, regular physical activity should be strongly promoted within lifestyle interventions for the prevention and treatment of dyslipidemia.

The US Preventive Services Task Force’s (USPSTF) identifies dyslipidemia as a CVD risk factor, defined as LDL cholesterol >130 mg/dL or HDL cholesterol <40 mg/dL. Therefore, driven by our initial stratification analysis that suggested independent associations on HDL cholesterol, we studied the joint effect between anthocyanin intake and physical activity on HDL cholesterol <40 mg/dL followed by stratification and interaction analyses. While the stratification analysis tested for the possible effect modification of one factor on the causal pathway of another factor, the interaction analysis tested the potential synergism between two independent causal pathways to produce an effect greater than the sum or multiplication of the two individual effects. The rationale for studying interactions is to better understand which two exposures share inter-related biological mechanisms that create an observable synergistic effect, identifying to whom would an intervention be most advantageous [24]. For the correct calculation and interpretation of RERI, the combinations of anthocyanin intake and physical activity were created by presenting the variables as risk factors and the reference category determined as the category with the lowest risk when considered jointly [25]. The double exposed category for low anthocyanin intake and low physical activity showed the strongest association with HDL cholesterol <40 mg/dL; however, neither measures of multiplicative nor additive interaction were statistically significant.

Although our analysis lacked statistical significance, our results suggest a possible joint effect greater than the sum of the individual exposures most likely due to shared underlying mechanisms. A recent intervention in healthy adult males suggested anthocyanin intake duration affects metabolic responses, including fat and carbohydrate oxidation, during moderate-intensity walking exercise. This may be attributed to an enhanced bioavailability of anthocyanins-derived metabolites involved in mechanisms of oxidation during physical activity [26]. The highest concentrations of dietary anthocyanins derive from elderberries, chokeberries, bilberries, raspberries, black currants, blackberries, and blueberries, among others. However, currently, little is known on the bioavailability of anthocyanins and the concentration in these foods may vary greatly due to influences, such as genetic, environmental, and agronomic factors, including light, temperature, humidity, fertilization, food processing, and storage conditions [27]. Future randomized controlled trials should consider a combination of assessing dietary anthocyanin intake by a validated FFQ combined with biomarker assessment.

Limitations of this study include a possible misclassification bias, due to the self-reported nature of the data and unknown anthocyanin supplementation among participants. Although it is uncertain whether supplementation use would over or underestimate the effect of anthocyanin-rich foods, evidence suggests anthocyanin supplementation improves antioxidative and anti-inflammatory capacity in a dose–response manner and a greater effect would be observed for both subgroups with low and high intakes of anthocyanin-rich foods [13]. Nevertheless, the probable use of anthocyanin supplementation is very low considering these supplements were not indicated in the open-ended question on the type of supplements habitually consumed. Although physical activity was reported using a validated questionnaire, self-reported physical activity has been demonstrated to be less predictive of CVD risk than objective accelerometer measurements [22,23]. Likewise, the validated semiquantitative FFQ did not include all common food sources of anthocyanins and some items captured more than one food with varying anthocyanin content [28]. Due to the predominately male prevalence of the firefighter profession, our results must be extrapolated to women with precaution. Moreover, this study population of Midwestern US career firefighters is not representative of the general population; however, biological plausibility should be the basis for generalizations in epidemiology [29,30]. In addition, because of the cross-sectional design of this study, we cannot infer causality from our results but rather can generate viable hypotheses for future studies.

Although we studied the specific effect of anthocyanin intake, this plant-based bioactive compound is consumed within the bigger context of diet. It is the synergy between foods and nutrients that defines diet quality; therefore, our findings should be considered within the context of multiple possible pathways by which an inappropriate diet could lead to the development of CVD [31]. Although multivariable-adjusted models controlled extensively for potential lifestyle predictors of lipid parameters, including diet; measured with a previously validated mMDS score, residual confounding cannot be completely eliminated [32]. Nonetheless, an appropriate diet, such as the Mediterranean diet, rich in polyphenols through the frequent and abundant consumption of fruits, vegetables, wine, and extra virgin olive oil, may be a practical recommendation for achieving a high anthocyanin intake [31].

Considering multivariable analyses require a large sample size, interaction analyses require an even greater sample size. Nevertheless, the power to detect interactions tends to be greater on the additive scale than the multiplicative scale [33,34]. Due to our relatively small sample size, the statistical power is a limitation of our study; nonetheless, the joint effect and stratification analysis still offer substantial insight on the effect modification and potential interaction between the given exposures. Future studies with a larger sample size and longitudinal design are warranted to further study this hypothesis, while limiting the possibility of reverse causality [33]. In line with our findings across a diverse selection of lipid parameters, future studies may also consider focusing specifically on measurements of total cholesterol and HDL cholesterol, which have proven sufficient to capture the lipid-associated risk in CVD prediction [35].

The USPSTF currently recommends offering adults who are overweight or obese and have additional CVD risk factors intensive behavioral counseling interventions to promote a healthful diet and physical activity; meanwhile, evidence among those without known risk factors suggests a positive but small benefit for the prevention of CVD [36,37]. In regards to the promotion of anthocyanin-rich diets for healthy lipid profiles, previous studies have demonstrated berries, as main sources of anthocyanins, have greater effects on lipid concentrations in obese/overweight individuals (BMI >25 kg/m^2^), individuals with cardiovascular risk factors, those ≥50 years, or who have metabolic syndrome compared to healthy individuals [11,14,38]. Future research in this line conducted among a large representative population with the inclusion of additive interaction analysis could be instrumental for identifying narrower CVD risk subgroups and offer greater efficacy among behavior change interventions for the promotion of ideal cardiovascular health.

## 4. Materials and Methods

### 4.1. Study Population

*Feeding America’s Bravest* is a cluster-randomized diet intervention trial that included 44 fire stations from the Indianapolis Fire Department (IFD) and 6 fire stations from Fishers (IN) Fire Department. The primary objective of this RCT was to compare a Mediterranean Diet Nutritional Intervention (MDNI) with multiple behavior change strategies: diet/lifestyle education, discounted access to key Mediterranean diet foods, electronic education platforms and reminders, with a Midwestern-style diet or “usual care” group with a cross-over study design over a 2-year period.

Although 486 persons were enrolled, only 265 participants completed the baseline lifestyle questionnaire between November 28, 2016 and April 16, 2018; participants with missing FFQ or biochemical assessment (n = 3) and participants whose energy intake exceeded predefined levels (men: 800–5000 kcal/d, women: 500–3500 kcal/d) (n = 13) were excluded, leaving 249 participants for evaluation.

Full informed consent was received from participants who met the eligibility criteria at the time of enrollment. In accordance with the Declaration of Helsinki, all potential participants were informed of their right to refuse to participate or to withdraw from the study at any time without retribution. The study protocol was approved by the Harvard Institutional Review Board (IRB16–10170) and is registered at Clinical Trials (NCT029441757) [39]. More details on this study’s objective, design, and methods have been previously published elsewhere [40].

### 4.2. Dietary Assessment

Dietary intake was assessed at baseline using a validated 131-item semi-quantitative FFQ, which reflected the previous year’s habitual intake, and a lifestyle questionnaire with additional dietary information, including a 13-item modified Mediterranean diet score (mMDS) [28,41]. Flavonoid subclasses were calculated as the habitual daily intake (mg/day), estimated using the US Department of Agriculture (USDA) flavonoid content of foods database, according to previously described methods [42,43]. Commonly consumed US dietary sources of anthocyanins include fruits, such as berries, blackcurrants, red grapes, plums, and cherries, as well as red wine, fruit juices, and some vegetables, such as radishes [44,45]. Anthocyanins are further classified into six subclasses: pelargonidin, cyanidin, delphinidin, peonidin, petunidin, and malvidin.

### 4.3. Physical Activity

Physical activity level was collected at baseline using a validated physical activity questionnaire administered within the lifestyle questionnaire [46]. On a scale from 0–7, participants were asked to identify the statement which option best described their habitual level of physical activity over the past month: (0) Avoid walking or exertion (e.g., always use elevator, drive whenever possible instead of walking, biking, or rollerblading); (1) walk for pleasure, routinely use stairs, occasionally exercise sufficiently to cause heavy breathing or perspiration; (2) 10 to 60 min per week; (3) over one hour per week; (4) run less than 1 mile per week or spend less than 30 min per week in comparable physical activity; (5) run 1 to 5 miles per week or spend 30 to 60 min per week in comparable physical activity; (6) run 5 to 10 miles per week or spend 1 to 3 h per week in comparable physical activity; and (7) run over 10 miles per week or spend over 3 h per week in comparable physical activity (Appendix A).

### 4.4. Outcome Assessment

Baseline lipid panels were collected in participants’ biochemical assessments from the fire department medical examinations at Public Safety Medical (PSM) clinics. Blood samples were collected after an overnight fast at baseline and at follow-up. Plasma and serum were collected in 15-mL specific tubes and were aliquoted, frozen at −80 °C, and stored. Blood lipid profiles were determined using standardized automated high-throughput enzymatic analyses, which achieved coefficients of variation of ≤3% for cholesterol and ≤5% for triglycerides, using a cholesterol assay kit and reagents Ref:7D62–21 and triglyceride assay kit and reagents Ref:7D74–21 by ARCHITECT c System, Abbott Laboratories, IL, USA. Baseline measures were gathered from the PSM electronic medical record database within the last year from enrollment in the study. The primary outcomes of this study were lipid profile measures, specifically TGs, total cholesterol, HDL cholesterol, LDL cholesterol, and ratios for LDL:HDL, TG:HDL, and total cholesterol:HDL.

### 4.5. Covariate Assessment

Information on sociodemographic characteristics, dietary intake, lifestyle habits, anthropometric measurements, and medical history were collected at baseline through in-person data collection, an online lifestyle questionnaire, or medical record after informed consent was given. BMI was calculated by dividing weight by height squared (kg/m^2^). Energy intake was calculated using the baseline FFQ. The mMDS score, described in Appendix A, was replicated based on Yang et al. and Sotos-Prieto et al. [28,41]. Participants with dyslipidemia, hypertension, or type 2 diabetes were identified if they had a previous diagnosis of these conditions or were being treated with lipid-lowering, antihypertensive, or antidiabetic medications, respectively, within the previous year to enrollment.

### 4.6. Statistical Analysis

A continuous variable for total anthocyanin intake was transformed into units of SDs using the standardization method to obtain a normal distribution. In addition, a dichotomous variable of total anthocyanins was created to define high and low anthocyanin intake using the median as the cut-off point; the median intake was equivalent to 19.14 mg/day. Physical activity was used as a continuous variable for each unit (level) increase. As a dichotomous variable, high physical activity was defined as regularly participating in heavy physical exercise, such as running or jogging, swimming, cycling, etc., or engaging in vigorous aerobic activity, such as tennis, basketball, or handball (levels 4–7), whereas low physical activity represented none to regular recreation or work requiring modest physical activity, such as golf, horseback riding, calisthenics, gymnastics, table tennis, bowling, weight lifting, and yard work (levels 0–3).

Baseline characteristics of participants were presented according to low and high categories of anthocyanin intake and physical activity. Quantitative values were expressed as mean ± SD and qualitative variables as a percentage. Statistical significance of between-group variation between low and high categories for each exposure were tested using Student’s t-test for quantitative variables and chi-squared test for qualitative variables.

To determine the contribution of each food source to the between-person variance of total anthocyanin intake, stepwise-selection regression analyses and nested least-squares linear regression models were conducted. The cumulative R^2^ indicates the proportion of variability with the addition of each source, whereas the change in cumulative R^2^ identifies each source’s contribution to the total variability of anthocyanin intake. Moreover, the contribution of anthocyanins from each food source was presented as a percentage of the total anthocyanin intake. Additionally, subclasses of anthocyanins were presented as percentages of the total anthocyanin intake.

Multivariable linear regression models were used to determine the extent to which each continuous exposure of anthocyanin intake and physical activity level predicted lipid profile measures. Beta coefficients were reported with 95% confidence intervals (CIs) and *p*-values presented for each adjusted model. To control for potential confounding, multivariable adjusted models included age (years), sex (M/F), BMI (kg/m^2^), total energy intake (kcal/d), mMDS (points), smoking status (never, current, or former), education level (technical school, some college, associate’s degree/Bachelor’s degree or higher), marital status (married/single), multivitamin use (yes/no), supplement use (yes/no), sleep (hours/day), prevalent hypertension, dyslipidemia, and type 2 diabetes (yes/no). Additionally, the independent multivariable linear regression models for anthocyanin intake were adjusted for physical activity level, total time spent sitting down (hours/week), and time spent in front of the television, computer and in the car (hours/week), whereas the fully adjusted models for physical activity were adjusted for anthocyanin intake. Total time sitting and sedentary behavior showed a correlation coefficient of 0.28, indicating these covariates measured different forms of inactivity. A sensitivity analysis considered additional exclusions for chronic diseases, women, and supplement use. 

To assess the potential effect modification between anthocyanin intake and physical activity on HDL cholesterol, we followed the recommendations by Knol and Vanderweele [47]. First, the prevalence within each subgroup was presented as a percentage and the joint effect of the four possible combinations of low and high exposures of anthocyanin intake and physical activity on HDL <40 mg/dL as relative risks, adjusted for age, sex, and total energy intake. Relative risks were calculated using generalized linear models with Poisson distribution and robust standard errors [48]. A stratification analysis tested effect modification by assessing each dichotomous exposure stratified by the other. This was followed by a comprehensive interaction analysis by applying both multiplicative and additive interaction analyses. Multiplicative interaction was tested by comparing age-, sex-, and energy-adjusted models with and without the interaction term, whereas the RERI was assessed on the additive scale [49].

All analyses were conducted with Stata version 14.0 (StataCorp, College Station, TX, USA). All *p*-values are two-sided and were considered statistically significant at *p* < 0.05.

## Figures and Tables

**Figure 1 molecules-25-04398-f001:**
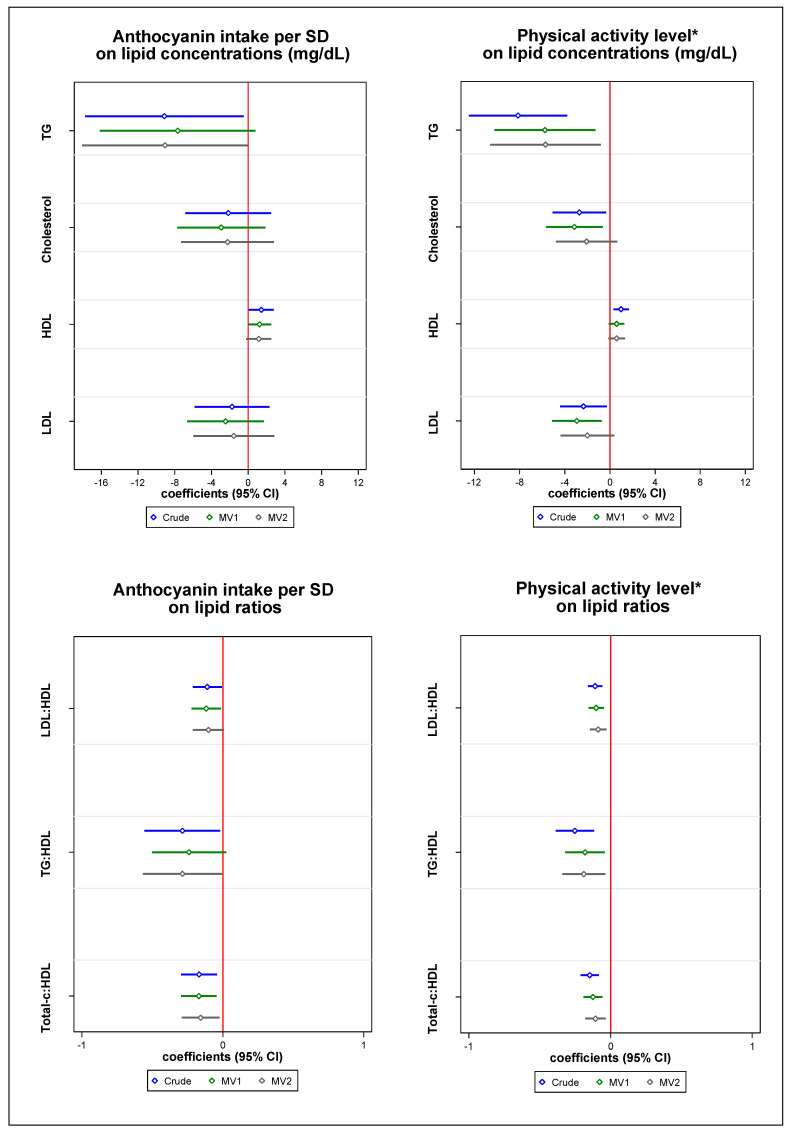
Independent associations (β, 95% CI) of anthocyanin intake (SD) and physical activity level with lipid concentrations (mg/dL) and lipid ratios. * Physical activity was assessed using a scale of 0–7 representing levels of physical activity ranging from none to running > 10 miles/wk or spending > 3 hrs/wk in comparable physical activity (Appendix A). MV1: multivariable model 1 adjusted for age, sex, BMI, smoking status, education level, marital status, prevalent hypertension, dyslipidemia, and type 2 diabetes. MV2: multivariable model 2 adjusted for age, sex, BMI, total energy intake, mMDS, smoking status, education level, marital status, multivitamin use, supplement use, sleep, prevalent hypertension, dyslipidemia, and type 2 diabetes. CI: confidence intervals, HDL: high density lipoprotein cholesterol, LDL: low density lipoprotein cholesterol, SD: standard deviation, TG: triglycerides, Total-c: total cholesterol. Appendix A shows β coefficients (95% CI) and *p*-vales of independent associations for all linear regression models.

**Table 1 molecules-25-04398-t001:** Baseline characteristics according to high and low anthocyanin intake and physical activity level.

	Anthocyanin Intake (SD) ^a^			Physical Activity Level ^b^		
	Low	High	*p*-Value	Low	High	*p*-Value
N	124	125		95	154	
Women (%)	4.8	5.6	0.79	2.11	7.14	0.08
Age (yrs)	47.2 (7.4)	46.3 (7.5)	0.30	48.7 (6.9)	45.5 (7.6)	0.002
BMI (kg/m^2^)	30 (4.5)	29.7 (4.2)	0.59	31.7 (4.4)	28.7 (3.9)	<0.001
Total energy intake (kcal/d)	2244 (941)	2491 (852)	0.03	2423 (993)	2334 (846)	0.45
mMDS ^†^ (pts)	22.3 (6.6)	25.3 (6.9)	<0.001	21 (7.6)	25.6 (5.8)	<0.001
Flavonoids (mg/d)	286 (212.0)	460 (282.0)	<0.001	383 (306.0)	367 (238.0)	0.65
Anthocyanins (mg/d)	10.9 (6.0)	53 (40.2)	<0.001	29.7 (35.5)	33.4 (35.7)	0.43
Protein intake (g/d)	97 (43.0)	110 (38.0)	0.01	104 (46.0)	103 (37.0)	0.95
Carbohydrate intake (g/d)	243 (106.0)	267 (99.0)	0.07	262 (106.0)	251 (101.0)	0.40
Whole grains (g/d)	33.2 (18.7)	39.9 (23.2)	0.01	35.7 (17.9)	37 (23.2)	0.64
Total fiber intake (g/d)	21 (8.7)	27.7 (9.2)	<0.001	23.7 (8.8)	24.8 (10.0)	0.39
Added sugar (g/d)	65.2 (48.7)	54.2 (35.2)	0.04	66.1 (47.7)	55.7 (39.0)	0.06
Fat intake (g/d)	93.5 (45.8)	101.4 (39.3)	0.09	101.2 (47.8)	95.1 (39.3)	0.27
Saturated fat (g/d)	31.4 (17.3)	31.8 (12.9)	0.85	33.3 (17.1)	30.6 (13.9)	0.18
Polyunsaturated fat (g/d)	19.4 (9.3)	22 (9.1)	0.02	21.6 (9.9)	20.2 (8.9)	0.27
Monounsaturated fat (g/d)	35.2 (17.6)	39.4 (16.8)	0.05	38.5 (18.8)	36.6 (16.3)	0.39
Alcohol (g/d)	9.1 (12.4)	15.6 (25.5)	0.01	11.2 (18.8)	13.1 (21.2)	0.48
Nondrinkers (%)	18.5	16.8	0.72	17.9	17.5	0.94
Smoking status (%)			0.29			0.54
never	57.3	52.8		54.7	55.2	
current	18.5	14.4		13.7	18.2	
former	24.2	32.8		31.6	26.6	
Education (%)			0.38			0.28
Technical school/some college/associates degree Bachelor’s degree or higher						
	66.9	61.6		68.4	61.7	
	33.1	38.4		31.6	38.3	
Marital status (%)			0.41			0.92
married	79.2	83.4		80	80.5	
single	21.8	17.6		20	19.5	
Multivitamin use (%)	38.7	38.4	0.96	31.6	42.9	0.08
Supplement use (proteins, glutamine, amino acids, etc.) (%)	24.2	39.2	0.01	20.0	39.0	0.002
Sitting (hrs/wk)	19.5 (13.0)	18.7 (16.8)	0.64	22.4 (19.4)	17.1 (11.1)	0.01
TV, computer, and driving (hrs/wk)	8.02 (3.8)	7.7 (4.0)	0.52	8.33 (4.4)	7.56 (3.5)	0.14
Sleep (hrs/d)	6.57 (1.1)	6.38 (0.9)	0.13	6.41 (1.1)	6.51 (1.0)	0.42
Prevalent hypertension (%)	5.65	6.4	0.80	7.37	5.19	0.48
Prevalent dyslipidemia (%)	20.2	7.2	0.003	10.5	15.6	0.26
Prevalent type 2 diabetes (%)	1.61	1.6	0.99	1.05	1.95	0.51

Values are means ± (SD), unless specified as a percentage (%). Percentages may not equal 100 due to rounding. ^a^ Low intake was defined as standard deviations of anthocyanin intake (mg/d) below the median and high intake above the median. ^b^ Low activity was defined as levels 0–3 representing those who avoid walking or exertion to >1 hr/wk of modest PA, whereas high activity levels 4–7 indicated running <1 mile/wk or spending <30 min/wk in heavy PA to running over 10 miles/wk or >3 hrs/wk of comparable PA. ^†^ An explanation of how this score was developed is shown in Appendix A. BMI: body mass index is weight in kilograms divided by meters squared, d: day, g: grams, hrs: hours, kcal: kilocalories, mg: milligrams, mMDS: modified Mediterranean Diet Score, N: population size, pts: points, TV: television, wk: week.

**Table 2 molecules-25-04398-t002:** Top contributors of total anthocyanin intake; source content, between-person variability, and contribution to total anthocyanin intake according to the FFQ items in Feeding America’s Bravest trial (2016–2019).

Sources (Serving Size)	mg/Serving	Cumulative R^2^	Change in R²	Contribution (%)
Blueberries (1/2 cup)	120.8	0.961	---	43
Strawberries (1/2 cup)	20.5	0.981	0.020	11
Red wine (5 oz. glass)	28.3	0.989	0.008	11
Apple or pears (1 fresh)	6.05 *	0.994	0.005	8
Raisins or grapes (1 oz or small pack) or (1/2 cup)	36.5	0.997	0.003	15

* anthocyanin content is an average of the anthocyanins in apples (8.4 mg/serv) and pears (3.7 mg/serv). Cumulative R^2^ indicates the proportion of variability with the addition of each source. The change in R^2^ indicates the between-person variability corresponding to each source.

**Table 3 molecules-25-04398-t003:** Association (β, 95% CI) between anthocyanin intake (independent variable in SD units) and different lipid parameters (mg/dL) stratified by subgroups of low and high physical activity level.

Lipid Profile	Anthocyanin Intake (Per SD)	
	Low PA * (n = 95)	High PA * (n = 154)
**Triglycerides**		
Age, sex, and energy adjusted model (95% CI)	−8.47 (−22.38 to 5.45)	−8.95 (−19.64 to 1.74)
Multivariable adjusted model 1 (95% CI) ^a^	−7.47 (−23.34 to 8.40)	−7.83 (−18.21 to 2.56)
Multivariable adjusted model 2 (95% CI) ^b^	−11.29 (−27.19 to 4.61)	−6.57 (−18.32 to 5.19)
**Total cholesterol**		
Age, sex, and energy adjusted model (95% CI)	−3.75 (−11.07 to 3.56)	−0.97 (−6.94 to 5.00)
Multivariable adjusted model 1 (95% CI) ^a^	−2.34 (−10.62 to 5.95)	−1.94 (−7.97 to 4.09)
Multivariable adjusted model 2 (95% CI) ^b^	−4.38 (−12.74 to 3.97)	2.80 (−3.79 to 9.39)
**HDL cholesterol**		
Age, sex, and energy adjusted model (95% CI)	0.52 (−1.51 to 2.55)	1.97 (0.25 to 3.69)
Multivariable adjusted model 1 (95% CI) ^a^	1.09 (−1.08 to 3.25)	1.55 (−0.07 to 3.17)
Multivariable adjusted model 2 (95% CI) ^b^	0.83 (−1.39 to 3.05)	1.73 (−0.10 to 3.57)
**LDL cholesterol**		
Age, sex, and energy adjusted model (95% CI)	−2.16 (−8.61 to 4.30)	−1.31 (−6.59 to 3.97)
Multivariable adjusted model 1 (95% CI) ^a^	−1.39 (−8.66 to 5.87)	−2.09 (−7.47 to 3.29)
Multivariable adjusted model 2 (95% CI) ^b^	−2.66 (−10.17 to 4.86)	2.16 (−3.72 to 8.04)
**LDL:HDL**		
Age, sex, and energy adjusted model (95% CI)	−0.10 (−0.28 to 0.08)	−0.11 (−0.23 to 0.01
Multivariable adjusted model 1 (95% CI) ^a^	−0.12 (−0.32 to 0.08)	−0.11 (−0.23 to 0.02)
Multivariable adjusted model 2 (95% CI) ^b^	−0.13 (−0.33 to 0.07)	−0.04 (−0.18 to 0.09)
**TG:HDL**		
Age, sex, and energy adjusted model (95% CI)	−0.22 (−0.69 to 0.25)	−0.31 (−0.62 to 0.00)
Multivariable adjusted model 1 (95% CI) ^a^	−0.23 (−0.75 to 0.30)	−0.27 (−0.57 to 0.03)
Multivariable adjusted model 2 (95% CI) ^b^	−0.31 (−0.84 to 0.22)	−0.28 (−0.62 to 0.07)
**Total cholesterol:HDL**		
Age, sex, and energy adjusted model (95% CI)	−0.16 (−0.37 to 0.06)	−0.17 (−0.32 to −0.01)
Multivariable adjusted model 1 (95% CI) ^a^	−019 (−0.42 to 0.05)	−0.15 (−0.30 to 0.00)
Multivariable adjusted model 2 (95% CI) ^b^	−0.21 (−0.44 to 0.03)	−0.09 (−0.26 to 0.08)

Coefficients for triglycerides, total cholesterol, HDL and LDL cholesterol show the strength of the effect on mg/dL of lipid concentration per SD of anthocyanin intake, whereas coefficients for LDL:HDL, TG:HDL, and total cholesterol:HDL show the effect on the ratio of lipid concentrations per SD of anthocyanin intake. * Low PA ranged from none to regular recreation or work requiring modest physical activity, such as golf, horseback riding, calisthenics, gymnastics, table tennis, bowling, weight lifting, yard work (levels 0–3); High PA was defined as regularly participating in heavy physical exercise such as running or jogging, swimming, cycling, rowing, skipping rope, running in place or engaging in vigorous aerobic activity such as tennis, basketball, or handball (levels 4–7). ^a^ Adjusted for age, sex, BMI, smoking status, education level, marital status, prevalent hypertension, dyslipidemia, and type 2 diabetes. ^b^ Adjusted for age, sex, BMI, total energy intake, mMDS, smoking status, education level, marital status, multivitamin use, supplement use, sleep, prevalent hypertension, dyslipidemia, and type 2 diabetes. CI: confidence intervals, HDL: high density lipoprotein cholesterol, LDL: low density lipoprotein cholesterol, PA: physical activity, SD: standard deviation, TG: triglycerides.

**Table 4 molecules-25-04398-t004:** Association (β, 95% CI) between physical activity (independent variable in level units) and different lipid parameters (mg/dL) stratified by subgroups of low and high anthocyanin intake.

Lipid Profile	Physical Activity Level (Per Unit)	
	Low Anthocyanin Intake * (n = 124)	High Anthocyanin Intake * (n = 125)
**Triglycerides**		
Age, sex, and energy adjusted model (95% CI)	−6.19 (−13.03 to 0.65)	−8.88 (−14.53 to −3.23)
Multivariable adjusted model 1 (95% CI) ^a^	−4.33 (−11.34 to 2.68)	−5.34 (−11.50 to 0.81)
Multivariable adjusted model 2 (95% CI) ^b^	−4.96 (−12.45 to 2.53)	−4.79 (−11.62 to 2.04)
**Total Cholesterol**		
Age, sex, and energy adjusted model (95% CI)	−2.69 (−6.01 to 0.64)	2.70 (−6.21 to 0.82)
Multivariable adjusted model 1 (95% CI) ^a^	−3.39 (−6.93 to 0.15)	−2.79 (−6.75 to 1.16)
Multivariable adjusted model 2 (95% CI) ^b^	−2.86 (−6.65 to 0.93)	−0.65 (−4.99 to 3.69)
**HDL Cholesterol**		
Age, sex, and energy adjusted model (95% CI)	0.82 (−0.15 to 1.80)	0.86 (−0.11 to 1.83)
Multivariable adjusted model 1 (95% CI) ^a^	0.73 (−0.29 to 1.75)	0.11 (−0.89 to 1.11)
Multivariable adjusted model 2 (95% CI) ^b^	0.65 (−0.44 to 1.74)	0.35 (−0.77 to 1.46)
**LDL Cholesterol**		
Age, sex, and energy adjusted model (95% CI)	−2.24 (−5.08 to 0.60)	−2.47 (−5.68 to 0.74)
Multivariable adjusted model 1 (95% CI) ^a^	−3.34 (−6.28 to −0.20)	−2.63 (−6.25 to 0.98)
Multivariable adjusted model 2 (95% CI) ^b^	−2.56 (−5.81 to 0.68)	−0.99 (−4.98 to 3.01)
**LDL:HDL**		
Age, sex, and energy adjusted model (95% CI)	−0.10 (−0.17 to −0.03)	−0.11 (−0.19 to −0.03)
Multivariable adjusted model 1 (95% CI) ^a^	−0.12 (−0.19 to −0.04)	−0.08 (−0.17 to 0.01)
Multivariable adjusted model 2 (95% CI) ^b^	−0.10 (−0.18 to −0.02)	−0.06 (−0.16 to 0.04)
**TG:HDL**		
Age, sex, and energy adjusted model (95% CI)	−0.18 (−0.38 to 0.02)	−0.29 (−0.48 to −0.09)
Multivariable adjusted model 1 (95% CI) ^a^	−0.14 (−3.35 to 0.07)	−0.17 (−0.38 to 0.035)
Multivariable adjusted model 2 (95% CI) ^b^	−0.15 (−0.37 to 0.06)	−0.18 (−0.41 to 0.05)
**Total cholesterol:HDL**		
Age, sex, and energy adjusted model (95% CI)	−0.14 (−0.23 to −0.05)	−0.14 (−0.23 to −0.04)
Multivariable adjusted model 1 (95% CI) ^a^	−0.15 (−0.24 to −0.05)	−0.08 (−0.18 to 0.02)
Multivariable adjusted model 2 (95% CI) ^b^	−0.13 (−0.23 to −0.03)	−0.06 (−0.18 to 0.05)

Coefficients for triglycerides, total cholesterol, HDL and LDL cholesterol show the strength of the effect on mg/dL of lipid concentration per unit of physical activity level, whereas coefficients for LDL:HDL, TG:HDL, and total cholesterol:HDL show the effect on the ratio of lipid concentrations per unit of physical activity level. * High and low anthocyanin intake were defined as standard deviations of anthocyanin intake above or below the median, respectively. ^a^ Adjusted for age, sex, BMI, smoking status, education level, marital status, prevalent hypertension, dyslipidemia, and type 2 diabetes. ^b^ Adjusted for age, sex, BMI, total energy intake, mMDS, smoking status, education level, marital status, multivitamin use, supplement use, sleep, prevalent hypertension, dyslipidemia, and type 2 diabetes. CI: confidence intervals, HDL: high density lipoprotein cholesterol, LDL: low density lipoprotein cholesterol, TG: triglycerides.

**Table 5 molecules-25-04398-t005:** Prevalence, joint effect relative risk (RR), stratification, and multiplicative and additive interactions of anthocyanin intake and physical activity on HDL cholesterol <40 mg/dL.

HDL Cholesterol < 40 mg/dL		Anthocyanin Intake (SD)				RR (95% CI) ^c^ for Anthocyanin Intake Stratified by Physical Activity
		High ^a^		Low ^b^		
		Prevalence (%)	RR (95% CI)^c^	Prevalence (%)	RR (95% CI) ^c^	
Physical Activity	High ^a^	12.66	1 Ref.	18.67	1.46 (0.68 to 3.11); *p* = 0.33	1.46 (0.67 to 3.18); *p* = 0.34
	Low ^b^	32.61	2.36 (1.15 to 4.83); *p* = 0.02	38.78	2.83 (1.42 to 5.67); *p* = 0.003	1.21 (0.70 to 2.08); *p* = 0.50
RR (95% CI) ^d^ for physical activity stratified by anthocyanin intake			2.19 (1.07 to 4.49); *p* = 0.03		1.99 (1.10 to 3.60); *p* = 0.02	
Measure of interaction on multiplicative scale: Ratio of RR (95% CI)					*p* = 0.72	
Measure of interaction on additive scale: RERI (95% CI)					0.02 (−1.63 to 1.66); *p* = 0.98	

Regressions are adjusted for age, sex, and energy intake. ^a^ High anthocyanin intake: standard deviations above the median anthocyanin intake; High PA: regularly participating in heavy physical exercise such as running or jogging, swimming, cycling, etc. or engaging in vigorous aerobic activity such as tennis, basketball, or handball (levels 4–7). ^b^ Low anthocyanin intake: standard deviations below the median anthocyanin intake; Low PA: none to regular recreation or work requiring modest physical activity, such as golf, horseback riding, calisthenics, gymnastics, table tennis, bowling, weightlifting, yard work (levels 0–3). ^c^ Relative risks represent low vs high anthocyanin intake stratified by physical activity level. ^d^ Relative risks represent low vs high physical activity stratified by anthocyanin intake. CI: confidence intervals, HDL: high density lipoprotein, RERI: relative excess risk due to interaction, RR: relative risk.

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
