# Peer review of "Anthocyanin Intake and Physical Activity: Associations with the Lipid Profile of a US Working Population"

_molecules, 2020, doi:10.3390/molecules25194398_

Round 1

Reviewer 1 Report

This was a cross-sectional analysis to explore the individual and interactive association of habitual anthocyanins intake and physical activity on lipid measures among 249 firefighters based on their baseline data of a clinical trial. The study novelty lies in that it is the first study to assess the joint exposures of anthocyanins intake and physical activity on lipids with both additive and multiplicative interaction being testified. However, the methods for evaluation of two exposures, statistical analysis and results description for interaction had some limitations which should be improved before consideration of publication.        My comments on the manuscript were listed below for consideration.

For methods

  • For estimation of anthocyanins intake by FFQ, it would be better to further describe: how many food items in FFQ contributed to the total anthocyanins amount? Whether anthocyanins exposure had been validated by certain biomarkers to indicate their bioavailability? A citation is needed in the corresponding text in methodology to indicate the validation of the FFQ used.
  • For outcome measures, although the information of lipids was obtained from medical records, it was still necessary to describe blood collection, methods for lipids testing, reagent kits used and inter- and intra- CVs as well.

For statistical analysis

  • Although this was a cross-sectional analysis, in order to avoid possibly reversal association, it would be better to exclude patients of chronic disorders or patients undertaking lipids lowering or other medications treatment since patients who were aware of the diagnosis of diseases may change their lifestyle or dietary habits. Sensitivity analysis might be still necessary even you have adjusted the variables of metabolic diseases in the regression models.
  • The non-significant findings of interaction between anthocyanins intake and physical activity on lipids could be due to inadequate power. Generally, multi-variable regression model requires more sample size due to adjustment of a range of covariates than that of univariate. Much more sample size (>4 folds) is needed for interactive testing than that of individual exposures. Multiplicative interaction requires more samples than that of additive interaction.
  • For statistical analysis, standardization could not make correction for skewed distribution. The comparison between two groups should be made by t-test but not ANOVA (quantitative) (Line 151-152.). For the multivariable regression model, I have concerns that the inclusion of MDS may lead to over-adjustment since notable correlation could be existed between anthocyanin intakes and MDS. In addition, only 5% participants were women, it may be better to further conduct sensitivity analysis among male alone.

For results and discussion

  • Table 1 indicated over half participants had supplements usage. Please describe clearly: what kind of supplements they were using? If the supplements included anthocyanins, thus the estimation of dietary sourced anthocyanins alone on lipids was far from adequate as supplementation dosages were much higher than those from dietary source. It could be better to use energy adjusted values for estimation of anthocyanins exposure. Table 1 also indicated that there were high prevalent of dyslipidemia and hypertension, then how about their medications treatment?
  • For testing of interaction, the manuscript only presented the results on HDL-c, how about the interactive effects on other lipids? Results presented in Table 5 had errors in calculation the measures of relative risks of one exposure stratified by another exposure. There was no value reported for measure of multiplicative interaction (only p value indicated in the footnote of Table 5). The common measures of interaction included not only RERI (for additive scale) but also synergy index (SI), attributable proportion (AP) or risk difference (RD). The study was cross-sectional design not case-control, so it could be better to report RR than OR for estimation of interaction.
  • For discussion, since the novelty of the analysis was the interaction of anthocyanins and physical activity, it would be better to include more discussion on the interaction and the underlying mechanisms. Line 354-56, the citation should be included for the RCT.
  • A number of grammar errors should be rectified.

Author Response

Response to Reviewer 1 Comments

Comments and Suggestions for Author

This was a cross-sectional analysis to explore the individual and interactive association of habitual anthocyanins intake and physical activity on lipid measures among 249 firefighters based on their baseline data of a clinical trial. The study novelty lies in that it is the first study to assess the joint exposures of anthocyanins intake and physical activity on lipids with both additive and multiplicative interaction being testified. However, the methods for evaluation of two exposures, statistical analysis and results description for interaction had some limitations which should be improved before consideration of publication.        My comments on the manuscript were listed below for consideration.

Thank you very much for taking the time to review our manuscript. We have fully addressed your suggestions/comments within the manuscript and have clarified various points to improve the description of our methods and present our results more clearly.

For methods

  • For estimation of anthocyanins intake by FFQ, it would be better to further describe: how many food items in FFQ contributed to the total anthocyanins amount?

Thank you for your comment on the methods for the estimation and validity of anthocyanin intake. We agree a full list of food sources included in the calculation of anthocyanin intake would be more complete, however, we indicate in Table 2 the top 5 contributors of anthocyanins (blueberries, strawberries, red wine, apple or pears, and raisins or grapes) accounted for 88% of total anthocyanins.

Existing text in results: “Blueberries were the richest source of anthocyanins contributing 43% to total anthocyanin intake (Table 2). Raisins and grapes represented the second richest source (15%); however they showed the least between-person variability. Strawberries, red wine, apples and pears followed in contribution and descending between-person variability, respectively. The composition of anthocyanin subclasses was as follows; 27% Malvidin, 24% Cyanidin, 18% Delphinidin, 17% Pelargonidin, 10% Petunidin, and 4% Peonidin, according to the above-mentioned anthocyanin-rich sources which accounted for 88% of total anthocyanins.”

These food items were therefore the main contributors to total anthocyanin intake, measured with the FFQ and derived using the US Department of Agriculture (USDA) flavonoid content of foods database, according to previously described methods (1,2). Other sources, including peaches or plums, apricots, and prunes or dried plums contributed small amounts to the total anthocyanin intake in our study. Additional sources of anthocyanins, which were not collected in the study FFQ include some red to purplish blue-colored vegetables and grains, such as purple corn, purple sweet potato, red cabbage, black carrot, black soybean, and some varieties of rice (3). The sum of the top 5 sources had a practically identical mean and range compared to the total anthocyanin variable (sum of all food components sources of anthocyanin) provided: 30.02 mg/d (0-156.94) (top 5 sources) compared to 31.16 mg/d (0.15-156.82) (all food sources).

Incorporated text in discussion: Our top 5 sources of anthocyanin rich foods included blueberries, strawberries, red wine, apple or pears, and raisins or grapes accounted for 88% of total anthocyanin intake. The remaining 12% can be attributed to other sources, including peaches or plums, apricots, bananas, cantaloupe, and prunes which contributed small amounts to the total anthocyanin intake in our study. Additional sources of anthocyanins, which were not collected in the study FFQ include some red to purplish blue-colored vegetables and grains, such as purple corn, purple sweet potato, red cabbage, black carrot, black soybean, and some varieties of rice (3).”

Whether anthocyanins exposure had been validated by certain biomarkers to indicate their bioavailability?

Unfortunately, the Feeding America’s Bravest trial does not have biomarkers associated with anthocyanin intake for further validation of the derived intake reported via the FFQ. We have included this within our study limitations and suggested future research considers integrating intake and biomarkers to more accurately assess anthocyanin intake.

Incorporated text in discussion: “The highest concentrations of dietary anthocyanins derive from elderberries, chokeberries, bilberries, raspberries, black currants, blackberries, and blueberries, among others. However, currently little is known on the bioavailability of anthocyanins and the concentration in these foods may vary greatly due to influences such as genetic, environmental, and agronomic factors, including light, temperature, humidity, fertilization, food processing, and storage conditions (4). Future randomized controlled trials should consider a combination of assessing dietary anthocyanin intake by a validated FFQ combined with biomarker assessment.”

A citation is needed in the corresponding text in methodology to indicate the validation of the FFQ used.

The validation of the semiquantitative FFQ is currently cited in methods as reference number 13 and has been additionally cited in the limitations section of the discussion:

Willett, W.C.; Sampson, L.; Stampfer, M.J.; Rosner, B.; Bain, C.; Witschi, J.; Hennekens, C.H.; Speizer, F.E. Reproducibility and Validity of a Semiquantitative Food Frequency Questionnaire. Am. J. Epidemiol., 1985, 122 (1), 51–65. https://doi.org/10.1093/oxfordjournals.aje.a114086.

  • For outcome measures, although the information of lipids was obtained from medical records, it was still necessary to describe blood collection, methods for lipids testing, reagent kits used and inter- and intra- CVs as well.

Thank you, we have included additional details specifying the methods used for lipid collection data.

Incorporated text in methods: Blood samples were collected after an overnight fast at baseline and at follow-up. Plasma and serum were collected in 15mL specific tubes and were aliquoted, frozen at −80◦C, and stored. Blood lipid profiles were determined using standardized automated high throughput enzymatic analyses, which achieved coefficients of variation of ≤3% for cholesterol and ≤5% for triglycerides, using cholesterol assay kit and reagents Ref:7D62-21 and triglyceride assay kit and reagents Ref:7D74-21 by ARCHITECT c System, Abbott Laboratories, IL, USA.

For statistical analysis

  • Although this was a cross-sectional analysis, in order to avoid possibly reversal association, it would be better to exclude patients of chronic disorders or patients undertaking lipids lowering or other medications treatment since patients who were aware of the diagnosis of diseases may change their lifestyle or dietary habits. Sensitivity analysis might be still necessary even you have adjusted the variables of metabolic diseases in the regression models.

Thank you, we have conducted an additional sensitivity analysis for the linear regressions of both exposures on lipid parameters (Supplementary table 4) excluding those participants with chronic disease, defined as a previous diagnosis or current treatment for hypertension, dyslipidemia, or type 2 diabetes. This led to the exclusion of 50 additional participants: hypertension (n=15), dyslipidemia (n=34), or type 2 diabetes (n=4), which did not implicate substantial changes from our primary results.

Supplementary Table 4: Sensitivity analysis for anthocyanin intake (SD) and physical activity level on lipid profile measures (mg/dL).

Sensitivity Analysis

Anthocyanin intake (SD)

Physical activity level*

Exclusions

n

β (95%CI)

p-value

β (95%CI)

p-value

Chronic diseases†a

199

Triglycerides

-9.09 (-19.24 to 1.06)

0.08

-5.89 (-11.64 to -0.14)

0.05

Total cholesterol

-2.29 (-7.67 to 3.12)

0.41

-0.55 (-3.62 to 2.52)

0.72

HDL cholesterol

0.92 (-0.59 to 2.44)

0.23

0.85 (-0.01 to 1.70)

0.05

LDL cholesterol

-1.44 (-6.14 to 3.27)

0.55

-0.83 (-3.50 to 1.84)

0.54

LDL:HDL

-0.09 (-0.21 to 0.03)

0.16

-0.08 (-0.15 to -0.01)

0.03

TG:HDL

-0.27 (-0.58 to 0.04)

0.09

-0.21 (-0.39 to -0.04)

0.02

Total cholesterol:HDL

-0.14 (-0.29 to 0.01)

0.07

-0.10 (-0.18 to -0.01)

0.03

Womena

236

Triglycerides

-9.32 (-18.81 to 0.16)

0.05

-5.69 (-10.76 to -0.63)

0.03

Total cholesterol

-2.10 (-7.25 to 3.04)

0.42

-1.97 (4.72 to 0.77)

0.16

HDL cholesterol

0.72 (-0.67 to 2.12)

0.31

0.64 (-0.11 to 1.38)

0.09

LDL cholesterol

-1.01 (-5.58 to 3.55)

0.66

-1.97 (-4.40 to 0.46)

0.11

LDL:HDL

-0.08 (-0.20 to 0.03)

0.16

-0.09 (-0.15 to -0.03)

0.004

TG:HDL

-0.29 (-0.58 to 0.01)

0.06

-0.19 (-0.35 to -0.03)

0.02

Total cholesterol:HDL

-0.14 (-0.28 to 0.00)

0.05

-0.11 (-0.18 to -0.03)

0.004

Supplement use (proteins, glutamine, amino acids, etc.)a

170

Triglycerides

-8.72 (-19.39 to 1.96)

0.11

-4.79 (-10.38 to 0.80)

0.09

Total cholesterol

-4.22 (-10.53 to 2.08)

0.19

-3.56 (-6.84 to -0.29)

0.03

HDL cholesterol

0.38 (-1.36 to 2.11)

0.69

0.23 (-0.68 to 1.14)

0.61

LDL cholesterol

-2.77 (-8.44 to 2.91)

0.34

-2.83 (-5.78 to 0.12)

0.06

LDL:HDL

-0.09 (-0.22 to 0.05)

0.21

-0.07 (-0.14 to -0.00)

0.04

TG:HDL

-0.21 (-0.51 to 0.10)

0.18

-0.10 (-0.26 to 0.06)

0.21

Total cholesterol:HDL

-0.13 (-0.29 to 0.03)

0.12

-0.09 (-0.18 to -0.01)

0.03

*Physical activity was assessed using a scale of 0-7 representing levels of physical activity ranging from none to running >10 miles/wk or spending >3 hrs/wk in comparable physical activity.

Chronic diseases was defined as reporting a previous diagnosis or current treatment for hypertension, dyslipidemia, or diabetes, respectively.

aAdjusted for age, sex, BMI, total energy intake, mMDS, smoking status, education level, marital status, multivitamin use, supplement use, sleep, anthocyanin intake, prevalent hypertension, dyslipidemia, and type 2 diabetes

TG: triglycerides, HDL: high density lipoprotein cholesterol, LDL: low density lipoprotein cholesterol

Boldface indicates statistical significance (p<0.05)

Incorporated text in methods: A sensitivity analysis considered additional exclusions for chronic diseases, women, and supplement use.

Incorporated text in results: Sensitivity analysis for both exposures with additional exclusions for chronic disease, women, and supplement use further supported the robustness of our findings (Table S4).

  • The non-significant findings of interaction between anthocyanins intake and physical activity on lipids could be due to inadequate power. Generally, multi-variable regression model requires more sample size due to adjustment of a range of covariates than that of univariate. Much more sample size (>4 folds) is needed for interactive testing than that of individual exposures. Multiplicative interaction requires more samples than that of additive interaction.

Thank you for your comment on the statistical power needed for interaction analysis, particularly on the multiplicative scale. We have added more text to cover this limitation of our study in the discussion and have cited Vanderweele on this topic of the sample size needed for these analyses. Moreover, we know that the cross-sectional design is useful to generate a new hypothesis and we have underlined in the text that further longitudinal studies with larger sample size are needed.

Incorporated text in discussion: Considering multivariable analyses require a large sample size, interaction analyses require even greater sample size. Nevertheless, the power to detect interactions tends to be greater on the additive scale than the multiplicative scale (5,6). Due to our relatively small sample size, the statistical power is a limitation of our study, nonetheless the joint effect and stratification analysis still offer substantial insight on the effect modification and potential interaction between the given exposure. Future studies with larger sample size and longitudinal design are warranted to further study this hypothesis, while limiting the possibility of reverse causality (5).

  • For statistical analysis, standardization could not make correction for skewed distribution. The comparison between two groups should be made by t-test but not ANOVA (quantitative) (Line 151-152.).

Thank you we have corrected the error for the comparison of quantitative variables between two groups and have properly indicated that the comparison between high and low exposures in Table 1 was conducted with Student’s t-test.

Modified text in methods: “Statistical significance of between-group variation between low and high categories for each exposure were tested using Student’s t-test for quantitative variables and chi-squared test for qualitative variables.”

For the multivariable regression model, I have concerns that the inclusion of MDS may lead to over-adjustment since notable correlation could be existed between anthocyanin intakes and MDS.

With regard to adjustment for an overall diet covariable, the mMDS was incorporated into the regression model to account for the possible associations between other dietary components on lipid profile that could interfere with the potential causal pathway between our exposures of anthocyanins and physical activity on lipid parameters. The fully adjusted models with mMDS seek to estimate the effect fully attributed to our exposures of interest. Nevertheless, we included Model 1, which does not include mMDS or covariables related with dietary intake, for the reason you have suggested.  Additionally, we calculated the correlation coefficient between the variables employed in our analysis for mMDS and total anthocyanin intake and observed a correlation coefficient of r=0.16, suggesting a very weak correlation.

In addition, only 5% participants were women, it may be better to further conduct sensitivity analysis among male alone.

Lastly, with regard to our 5% female population, we have included a sensitivity analysis for multivariable adjusted linear regressions with the exclusion of females (Supplementary Table 4). Additionally, below we have provided a comparison of males and females, by excluding the opposite sex, in a less adjusted model that allowed conducting regressions on lipid measures with the 13 female participants.

Incorporated text in methods: “A sensitivity analysis was conducted with the additional exclusion of chronic diseases, women, and supplement use.”

Incorporated text in results: Sensitivity analysis for both exposures with additional exclusions for chronic disease, women, and supplement use further supported the robustness of our findings (Table S4).

  • See Supplementary Table 4 above

Due to the statistical power among 13 female participants, the model here was the least adjusted multivariable model that adjusted for age, BMI, smoking status, education level, marital status, prevalent hypertension, dyslipidemia, and type 2 diabetes. We did not include this analysis in our findings due to the low statistical power for females and similar findings are presented in Supplementary Table 4 in a fully adjusted model with the additional exclusion of females.

Sensitivity Analysis

Anthocyanin intake (SD)

Physical activity level*

n

β (95%CI)

p-value

β (95%CI)

p-value

Mena

236

Triglycerides

-7.87 (-16.78 to 1.03)

0.08

-5.68 (-10.31 to -1.04)

0.02

Total cholesterol

-2.82 (-7.72 to 2.07)

0.26

-3.03 (-05.57 to -0.49)

0.02

HDL cholesterol

0.83 (-0.48 to 2.15)

0.22

0.62 (-0.06 to 1.31)

0.07

LDL cholesterol

-1.96 (-6.28 to 2.37)

0.37

-2.87 (-5.11 to -0.64)

0.01

LDL:HDL

-0.10 (-0.21 to 0.01)

0.07

-0.10 (-0.16 to -0.05)

<0.001

TG:HDL

-0.24 (-0.51 to 0.04)

0.09

-0.18 (-0.32 -0.04)

0.02

Total cholesterol:HDL

-0.15 (-0.28 to -0.02)

0.02

-0.12 (-0.01 to -0.06)

<0.001

Womena

13

Triglycerides

-6.15 (-27.75 to 15.45)

0.54

-11.57 (-36.74 to 13.59)

0.24

Total cholesterol

-34.78 (-54.18 to -15.39)

0.01

-12.85 (-37.91 to 12.21)

0.20

HDL cholesterol

0.42 (-22.2 to 23.04)

0.96

-0.19 (-11.85 to 11.47)

0.96

LDL cholesterol

-31.56 (-57.03 to -6.13)

0.03

-10.22 (-36.94 to 16.49)

0.31

LDL:HDL

-0.53 (-1.59 to 0.53)

0.21

-0.18 (-0.84 to 0.47)

0.44

TG:HDL

-0.29 (-1.79 to 1.21)

0.58

-0.23 (-0.93 to 0.46)

0.36

Total cholesterol:HDL

-0.59 (-1.87 to 0.69)

0.24

-0.23 (-0.98 to 0.52)

0.40

*Physical activity was assessed using a scale of 0-7 representing levels of physical activity ranging from none to running >10 miles/wk or spending >3 hrs/wk in comparable physical activity.

Chronic diseases was defined as reporting a previous diagnosis or current treatment for hypertension, dyslipidemia, or diabetes, respectively.

aAdjusted for age, BMI, smoking status, education level, marital status, prevalent hypertension, dyslipidemia, and type 2 diabetes.

TG: triglycerides, HDL: high density lipoprotein cholesterol, LDL: low density lipoprotein cholesterol

Boldface indicates statistical significance (p<0.05)

We have considered this characteristic of the firefighter working population with a very low percentage of females as a limitation of our study.

Modified text in discussion: Due to the predominately male prevalence of the firefighter profession, our results must be extrapolated to women with precaution. Moreover, this study population of Midwestern US career firefighters is not representative of the general population, however biological plausibility should be the basis for generalizations in epidemiology (7,8).

Modified text in discussion: “Future research in this line conducted among a large representative population with the inclusion of additive interaction analysis could be instrumental for identifying narrower CVD risk subgroups and offer greater efficacy among behavior change interventions for the promotion of ideal cardiovascular health.”

For results and discussion

  • Table 1 indicated over half participants had supplements usage. Please describe clearly: what kind of supplements they were using?

Thank you for bringing this to our attention. The questions related to supplement use included “Do you currently take supplements (proteins, glutamine, amino acids, etc.)?” “How many supplements do you take per week?”Please list the supplements that you take.”

We used the first question as a dichotomous variable, yes/no currently taking supplements. Table 1 now better reflects the self-reported information on supplement use by stating the questionnaire’s item more explicitly:

Modified text Table 1: Supplement use (proteins, glutamine, amino acids, etc.) (%). Vitamin use was also corrected to Multivitamin use.”

If the supplements included anthocyanins, thus the estimation of dietary sourced anthocyanins alone on lipids was far from adequate as supplementation dosages were much higher than those from dietary source. It could be better to use energy adjusted values for estimation of anthocyanins exposure.

With regard to the type of supplements, we observed supplements indicated in the open-ended question primarily specified protein or amino acids, multivitamins, and fish oil supplements. We do not have evidence to suspect that these kinds of supplements are contributing to the anthocyanin intake, considering anthocyanins are plant-based polyphenols. In addition, the dichotomous variable employed reflected the first question which specified (proteins, glutamine, amino acids, etc.), providing greater certainty that the participants were responding yes or no to protein/sports supplementation, which are not commonly antioxidant supplements. Nevertheless, we have acknowledged this possible misclassification bias of the exposure as a limitation of our study. Moreover, we adjusted our analyses for supplement use, as well as multivitamin use (yes/no).

Incorporated text: “Limitations of this study include a possible differential misclassification bias, due to the self-reported nature of the data and unknown anthocyanin supplementation among participants. Although it is uncertain whether supplementation use would over or underestimate the effect of anthocyanin-rich foods, evidence suggests anthocyanin supplementation improves anti-oxidative and anti-inflammatory capacity in a dose-response manner and a greater effect would be observed for both subgroups with low and high intakes of anthocyanin-rich foods (9). Nevertheless, the probable use of anthocyanin supplementation is very low considering these supplements were not indicated in the open-ended question that inquired on the type of supplements habitually consumed.”

Table 1 also indicated that there were high prevalent of dyslipidemia and hypertension, then how about their medications treatment?

Our definition of dyslipidemia included prior diagnosis: hypertension (n=6), type 2 diabetes (n=2), dyslipidemia (n=25) or current treatment: antihypertensives (n=12), lipid lowering medications (n=12), antidiabetics (n=2), therefore, our multivariable adjusted models adjusted for all possible lipid lowering, diabetes, and hypertension medications presented in Table 1, which could possibly be associated with total anthocyanin intake and metabolism.

In addition, we have conducted sensitivity analyses excluding those participants who reported using dietary supplements. This excluded an additional 79 participants, leaving a total of 170 participants for inclusion in this sensitivity analysis.

Incorporated text in methods: “A sensitivity analysis was conducted with the additional exclusion of chronic diseases, women, and supplement use.”

Incorporated text in results: “Sensitivity analysis for both exposures with additional exclusions for chronic disease, women, and supplement use further supported the robustness of our findings (Table S4).”

  • See Supplementary Table 4 above
  • For testing of interaction, the manuscript only presented the results on HDL-c, how about the interactive effects on other lipids?

Thank you for your comment, our selection of HDL cholesterol for the interaction analysis was based on the primary findings. Independently, the age, sex, and energy adjusted model showed statistically significant associations for anthocyanin intake β=1.43 (0.09 to 2.76) p=0.04 and physical activity β=0.99 (0.31 to 1.67) p=0.01 on HDL cholesterol. Although not statistically significant, the stratified analyses suggested a possible synergism between anthocyanin intake and physical activity on HDL cholesterol; anthocyanin intake across low physical activity β=0.83 (-1.39 to 3.05) vs high physical activity β =1.73 (-0.10 to 3.57) and physical activity among low anthocyanin intake β=0.65 (-0.44 to 1.74) vs high anthocyanin intake β=0.35 (-0.77 to 1.46) showed differences in effect across subgroups. This observation was not as clear for the other outcomes, possibly due to the low statistical power driven by the low sample size, which has been mentioned in our limitations for future studies to consider using larger longitudinal studies.

Existing text in discussion: Future studies with larger sample size and longitudinal design are warranted to further study this hypothesis, while limiting the possibility of reverse causality (5).

We also previously noted that total cholesterol and HDL cholesterol have been suggested to be sufficient predictors of CVD risk via lipid profiles.

Existing text in discussion: “In line with our findings across a diverse selection of lipid parameters, future studies may also consider focusing specifically on measurements of total cholesterol and HDL cholesterol, which have proven sufficient to capture the lipid-associated risk in CVD prediction (10).”

The common measures of interaction included not only RERI (for additive scale) but also synergy index (SI), attributable proportion (AP) or risk difference (RD).

With regards to the use of odds ratio, according to Vanderweele, when the probability of the outcome is rare in all exposure strata then odds ratios will approximate risk ratios (5). We previously indicated within our limitations the prevalence of cases ranged between 19 and 39% across subgroups, with a total of 58 cases within our study population (high-high: 10; low antho-high PA: 14; high antho-low PA: 15; low-low: 19). With this considered, the use of odds ratio was driven by the aim of this study which was to propose a hypothesis for longitudinal studies with stronger statistical power and follow-up that can deduce greater causal inference on the posible interaction between anthocyanin intake and physical activity on lipid parameters, as indicators of cardiovascular disease risk. These studies would then allow for studying absolute measures of impact, such as the risk difference, joint effect attributable proportions, or the population attributable fraction, associated with the combined risk as you suggested.

The study was cross-sectional design not case-control, so it could be better to report RR than OR for estimation of interaction.

We do agree that the OR has the limitation of overestimating the relative risk when the outcome is common and statistical power is low. Additionally, presenting OR may result in the misinterpretation of our findings if the magnitude of effect is interpreted as a relative risk, rather than the odds of one group compared to another group. Following your suggestion, and despite our original intention for using OR to focus on the measures of interaction, mainly RERI, we have presented the RR for all values in Table 5 for better accuracy and interpretation of our study’s findings on the possible interaction between anthocyanin intake and physical activity. The synergy index is another measure of additive interaction that employs risk ratios or odds ratios, however Vanderweele suggests the use of RERI, with covariate control, because the synergy index becomes difficult to interpret when one or both of the exposures is preventative rather than causative, such is our case (11).

Incorporated text in methods: “Relative risks were calculated using generalized linear mixed models with Poisson distribution and robust confidence intervals.”

Incorporated table replacing the existing Table 5:

Table 5. Prevalence, joint effect relative risk (RR), stratification, and multiplicative and additive interactions of anthocyanin intake and physical activity on HDL cholesterol<40mg/dL.

HDL cholesterol <40 mg/dL

Anthocyanin intake (SD)

RR (95%CI)c for anthocyanin intake stratified by physical activity

Higha

Lowb

Prevalence (%)

RR (95%CI)c

Prevalence (%)

RR (95%CI)c

Physical Activity

Higha

12.66

1 Ref.

18.67

1.46 (0.68 to 3.11);

p=0.33

1.46 (0.67 to 3.18); p=0.34

Lowb

32.61

2.36 (1.15 to 4.83);

p=0.02

  38.78

2.83 (1.42 to 5.67); p=0.003

1.21 (0.70 to 2.08);

p=0.50

RR (95%CI)d for physical activity stratified by anthocyanin intake

2.19 (1.07 to 4.49);

p=0.03

1.99 (1.10 to 3.60)

p=0.02

Measure of interaction on multiplicative scale: Ratio of RR (95%CI)

p=0.72

Measure of interaction on additive scale: RERI (95%CI)

0.02 (-1.63 to 1.66); p=0.98

Regressions are adjusted for age, sex, and energy intake.

aHigh anthocyanin intake: standard deviations above the median anthocyanin intake; High PA: regularly participating in heavy physical exercise such as running or jogging, swimming, cycling, etc. or engaging in vigorous aerobic activity such as tennis, basketball, or handball (levels 4-7).

bLow anthocyanin intake: standard deviations below the median anthocyanin intake; Low PA: none to regular recreation or work requiring modest physical activity, such as golf, horseback riding, calisthenics, gymnastics, table tennis, bowling, weight lifting, yard work (levels 0-3).

cregressions represent low vs high anthocyanin intake stratified by physical activity level.

dregressions represent low vs high physical activity stratified by anthocyanin intake.

CI: confidence intervals, RR: relative risk, CI: confidence intervals, RERI: relative excess risk due to interaction.

*Boldface indicates statistical significance.

There was no value reported for measure of multiplicative interaction (only p value indicated in the footnote of Table 5).

The habitual value presented for multiplicative interaction is the p-value for statistical significance of the comparison of the regression model with and without the product term. This is also the value proposed by Vanderweele in his recommendations for presenting analyses of effect modification and interaction (12). The product term’s odds ratio observed in the RR model presented in Table 5 is 0.82 (0.29 to 2.38), which corresponds with the p-value=0.72 presented in Table 5.

Results presented in Table 5 had errors in calculation the measures of relative risks of one exposure stratified by another exposure.

Thank you very much for highlighting the error in the stratification analysis, although we have replaced the OR table with RR analysis, we have reconducted the stratification analysis and corrected the values in Table 5 accordingly. The results for stratification now demonstrate the effect of low vs high anthocyanin intake stratified by high and low physical activity in one direction and low vs high physical activity stratified by high and low anthocyanin intake in the other direction. We have specified this in the new RR table footnotes for further clarification.

Table 5. Prevalence, joint effect relative risk (RR), stratification, and multiplicative and additive interactions of anthocyanin intake and physical activity on HDL cholesterol<40mg/dL.

HDL cholesterol <40 mg/dL

Anthocyanin intake (SD)

RR (95%CI)c for anthocyanin intake stratified by physical activity

Higha

Lowb

Prevalence (%)

RR (95%CI)c

Prevalence (%)

RR (95%CI)c

Physical Activity

Higha

12.66

1 Ref.

18.67

1.46 (0.68 to 3.11);

p=0.33

1.46 (0.67 to 3.18); p=0.34

Lowb

32.61

2.36 (1.15 to 4.83);

p=0.02

  38.78

2.83 (1.42 to 5.67); p=0.003

1.21 (0.70 to 2.08);

p=0.50

RR (95%CI)d for physical activity stratified by anthocyanin intake

2.19 (1.07 to 4.49);

p=0.03

1.99 (1.10 to 3.60)

p=0.02

Measure of interaction on multiplicative scale: Ratio of OR (95%CI)

p=0.72

Measure of interaction on additive scale: RERI (95%CI)

0.02 (-1.63 to 1.66); p=0.98

Regressions are adjusted for age, sex, and energy intake.

aHigh anthocyanin intake: standard deviations above the median anthocyanin intake; High PA: regularly participating in heavy physical exercise such as running or jogging, swimming, cycling, etc. or engaging in vigorous aerobic activity such as tennis, basketball, or handball (levels 4-7).

bLow anthocyanin intake: standard deviations below the median anthocyanin intake; Low PA: none to regular recreation or work requiring modest physical activity, such as golf, horseback riding, calisthenics, gymnastics, table tennis, bowling, weight lifting, yard work (levels 0-3).

cregressions represent low vs high anthocyanin intake stratified by physical activity level.

dregressions represent low vs high physical activity stratified by anthocyanin intake.

CI: confidence intervals, OR: odds ratio, CI: confidence intervals, RERI: relative excess risk due to interaction.

*Boldface indicates statistical significance.

  • For discussion, since the novelty of the analysis was the interaction of anthocyanins and physical activity, it would be better to include more discussion on the interaction and the underlying mechanisms.

We would like to respond to your suggestion by including the following text in the discussion.

Incorporated text in discussion: “Although our analysis lacked statistical significance, our results suggest a possible joint effect greater than the sum of the individual exposures most likely due to shared underlying mechanisms. A recent intervention in healthy adult males suggested anthocyanin intake duration affects metabolic responses, including fat and carbohydrate oxidation, during moderate intensity walking exercise. This may be attributed to an enhanced bioavailability of anthocyanins-derived metabolites involved in mechanisms of oxidation during physical activity (13). The highest concentrations of dietary anthocyanins derive from elderberries, chokeberries, bilberries, raspberries, black currants, blackberries, and blueberries, among others. However, currently little is known on the bioavailability of anthocyanins and the concentration in these foods may vary greatly due to influences such as genetic, environmental, and agronomic factors, including light, temperature, humidity, fertilization, food processing, and storage conditions (4). Future randomized controlled trials should consider a combination of assessing dietary anthocyanin intake by a validated FFQ combined with biomarker assessment.”

Line 354-56, the citation should be included for the RCT.

Thank you, the RCT has been cited at the end of the sentence.

Incorporated citation:

Kraus, W. E.; Houmard, J. A.; Duscha, B. D.; Knetzger, K. J.; Wharton, M. B.; McCartney, J. S.; Bales, C. W.; Henes, S.; Samsa, G. P.; Otvos, J. D.; et al. Effects of the Amount and Intensity of Exercise on Plasma Lipoproteins. N. Engl. J. Med., 2002, 347 (19), 1483–1492. https://doi.org/10.1056/NEJMoa020194.

  • A number of grammar errors should be rectified.

Thank you, the manuscript has been proofread and edited for grammar mistakes.

REFERENCES:

  1. Cassidy A, O’Reilly ÉJ, Kay C, Sampson L, Franz M, Forman JP, et al. Habitual intake of flavonoid subclasses and incident hypertension in adults. Am J Clin Nutr. 2011 Feb 1;93(2):338–47.
  2. Bhagwat S, Haytowitz D, Holden J. USDA Database for the Flavonoid Content of Selected Foods, release 3.1. U.S. Department of Agriculture. http//www ars usda gov/Services/docs htm?docid=6231. 2013;
  3. Khoo HE, Azlan A, Tang ST, Lim SM. Anthocyanidins and anthocyanins: Colored pigments as food, pharmaceutical ingredients, and the potential health benefits [Internet]. Vol. 61, Food and Nutrition Research. Swedish Nutrition Foundation; 2017 [cited 2020 Jun 22]. Available from: /pmc/articles/PMC5613902/?report=abstract
  4. Krga I, Milenkovic D. Anthocyanins: From Sources and Bioavailability to Cardiovascular-Health Benefits and Molecular Mechanisms of Action. J Agric Food Chem. 2019;
  5. Vanderweele TJ. Sample Size and Power Calculations for Additive Interactions [Internet]. Vol. 1, Epidemiologic Methods. Walter de Gruyter GmbH; 2012 [cited 2020 Jul 16]. p. 158–88. Available from: https://www.degruyter.com/view/journals/em/1/1/article-p159.xml
  6. Vanderweele T. Power and Sample-Size Claculations for Interaction Analysis. In: Explanation in Causal Inference: Methods for Mediation and Interaction. New York, NY: Oxford University Press; 2015. p. 346–68.
  7. Rothman K, Greenland S, Lash T. Modern Epidemiology. 3rd ed. Philadelphia, PA: Lippincott Williams and Wilkins; 2008.
  8. Rothman KJ, Gallacher JEJ, Hatch EE. Why representativeness should be avoided. Int J Epidemiol. 2013 Aug;42(4):1012–4.
  9. Zhang H, Xu Z, Zhao H, Wang X, Pang J, Li Q, et al. Anthocyanin supplementation improves anti-oxidative and anti-inflammatory capacity in a dose–response manner in subjects with dyslipidemia. Redox Biol. 2020 May 1;32:101474.
  10. Welsh C, Celis-Morales CA, Brown R, MacKay DF, Lewsey J, Mark PB, et al. Comparison of conventional lipoprotein tests and apolipoproteins in the prediction of cardiovascular disease data from UK biobank. Circulation [Internet]. 2019 Aug 13 [cited 2020 Sep 18];140(7):542–52. Available from: https://pubmed.ncbi.nlm.nih.gov/31216866/
  11. VanderWeele TJ. An Introduction to Interaction Analysis. In: Oxford University Press, editor. Explanation in Causal Inference: Methods for Mediation and Interaction [Internet]. New York, NY; 2015 [cited 2018 Aug 17]. p. 255–7. Available from: http://link.springer.com/10.1007/s10654-016-0189-8
  12. Knol MJ, VanderWeele TJ. Recommendations for presenting analyses of effect modification and interaction. Int J Epidemiol [Internet]. 2012 Apr 1 [cited 2018 May 7];41(2):514–20. Available from: https://academic.oup.com/ije/article-lookup/doi/10.1093/ije/dyr218

13.        Şahin, PhD MA, Bilgiç, PhD P, Montanari, MSc S, Willems, PhD MET. Intake Duration of Anthocyanin-Rich New Zealand Blackcurrant Extract Affects Metabolic Responses during Moderate Intensity Walking Exercise in Adult Males. J Diet Suppl [Internet]. 2020 [cited 2020 Sep 7]; Available from: https://www.tandfonline.com/doi/abs/10.1080/19390211.2020.1783421

Reviewer 2 Report

The manuscript entitled " Anthocyanin Intake and Physical Activity: Associations with the Lipid Profile of a US Working Population" is well written manuscript, describes the synergistic activity of anthocyanin intake and physical activity on lipid profile. 

Authors should describe the bioavailability of anthocyanin after intake through some reference studies. 

Author Response

Response to Reviewer 2 Comments

Open Review

English language and style

( ) Extensive editing of English language and style required 
( ) Moderate English changes required 
(x) English language and style are fine/minor spell check required 
( ) I don't feel qualified to judge about the English language and style 

Yes

Can be improved

Must be improved

Not applicable

Does the introduction provide sufficient background and include all relevant references?

(x)

( )

( )

( )

Is the research design appropriate?

(x)

( )

( )

( )

Are the methods adequately described?

(x)

( )

( )

( )

Are the results clearly presented?

( )

( )

( )

( )

Are the conclusions supported by the results?

(x)

( )

( )

( )

Comments and Suggestions for Authors

The manuscript entitled " Anthocyanin Intake and Physical Activity: Associations with the Lipid Profile of a US Working Population" is well written manuscript, describes the synergistic activity of anthocyanin intake and physical activity on lipid profile. 

Thank you very much for having taken the time to review our manuscript. We would like to resolve your concern on the bioavailability of dietary anthocyanins.

Authors should describe the bioavailability of anthocyanin after intake through some reference studies. 

We would like to respond to your suggestion by including the following text in the discussion.

Incorporated text in discussion: “Although our analysis lacked statistical significance, our results suggest a possible joint effect greater than the sum of the individual exposures most likely due to shared underlying mechanisms. A recent intervention in healthy adult males suggested anthocyanin intake duration affects metabolic responses, including fat and carbohydrate oxidation, during moderate intensity walking exercise. This may be attributed to an enhanced bioavailability of anthocyanins-derived metabolites involved in mechanisms of oxidation during physical activity (13). The highest concentrations of dietary anthocyanins derive from elderberries, chokeberries, bilberries, raspberries, black currants, blackberries, and blueberries, among others. However, currently little is known on the bioavailability of anthocyanins and the concentration in these foods may vary greatly due to influences such as genetic, environmental, and agronomic factors, including light, temperature, humidity, fertilization, food processing, and storage conditions (4). Future randomized controlled trials should consider a combination of assessing dietary anthocyanin intake by a validated FFQ combined with biomarker assessment.”

REFERENCES:

  1. Şahin, PhD MA, Bilgiç, PhD P, Montanari, MSc S, Willems, PhD MET. Intake Duration of Anthocyanin-Rich New Zealand Blackcurrant Extract Affects Metabolic Responses during Moderate Intensity Walking Exercise in Adult Males. J Diet Suppl [Internet]. 2020 [cited 2020 Sep 7]; Available from: https://www.tandfonline.com/doi/abs/10.1080/19390211.2020.1783421
  2. Krga I, Milenkovic D. Anthocyanins: From Sources and Bioavailability to Cardiovascular-Health Benefits and Molecular Mechanisms of Action. J Agric Food Chem. 2019;

Reviewer 3 Report

I do not think that the names of polyphenol and anthocyanin classes such as flavonols need to be capitalized since they are not proper nouns.

The sentence on lines 88-92 could be confusing. Could restate the sentence to something like " Although 486 persons were enrolled, only 265 participants completed the baseline lifestyle questionnaire between November 28, 2016 and April 16, 2018; participants with missing FFQ or biochemical assessment (n = 3) and participants whose energy intake exceeded predefined levels (men: 800-5000 kcal/d, women: 500-3500 kcal/d) (n = 13) were excluded, leaving 249 participants for evaluation.

Malvidin is misspelled on line 108.

What was the median anthocyanin intake used to create the dichotomous variable (lines 141-142)? This value should be mentioned in the text.

Please define mMDS in the methods.

Was there a reason that women  were included in the analyses despite the small sample size?

Author Response

Response to Reviewer 3 Comments

Open Review

English language and style

( ) Extensive editing of English language and style required 
( ) Moderate English changes required 
(x) English language and style are fine/minor spell check required 
( ) I don't feel qualified to judge about the English language and style 

Yes

Can be improved

Must be improved

Not applicable

Does the introduction provide sufficient background and include all relevant references?

( )

(x)

( )

( )

Is the research design appropriate?

(x)

( )

( )

( )

Are the methods adequately described?

( )

( )

(x)

( )

Are the results clearly presented?

(x)

( )

( )

( )

Are the conclusions supported by the results?

(x)

( )

( )

( )

I do not think that the names of polyphenol and anthocyanin classes such as flavonols need to be capitalized since they are not proper nouns.

Thank you, we have reviewed and corrected all names so that they do not appear capitalized.

Modified text in introduction: “Polyphenols are secondary plant metabolites and bioactive compounds naturally occurring in plants and plant-derived products, which can be differentiated into six main classes: flavones, flavonols, flavanols, flavanones, anthocyanins, and flavan-3-ols.”

The sentence on lines 88-92 could be confusing. Could restate the sentence to something like " Although 486 persons were enrolled, only 265 participants completed the baseline lifestyle questionnaire between November 28, 2016 and April 16, 2018; participants with missing FFQ or biochemical assessment (n = 3) and participants whose energy intake exceeded predefined levels (men: 800-5000 kcal/d, women: 500-3500 kcal/d) (n = 13) were excluded, leaving 249 participants for evaluation.

Thank you for your suggestion, we have replaced our sentence with your clearer and more concise sentence in methods.

Malvidin is misspelled on line 108.

Thank you for spotting this error, we have corrected the spelling of this subclass, as well as corrected the incorrect capitalization of each subclass.

Modified text in methods: “Anthocyanins are further classified into six subclasses: pelargonidin, cyanidin, delphinidin, peonidin, petunidin, and malvidin.”

What was the median anthocyanin intake used to create the dichotomous variable (lines 141-142)? This value should be mentioned in the text.

The median for anthocyanin intake in our study population was 19.14 mg/d which corresponds with the cut-off point used to create the dichotomous variable after the variable was transformed into units of standard deviation. This transformation was necessary to obtain a normal distribution of the exposure variable. We have included this value in methods  under statistical analysis as suggested. 

Incorporated text in methods: “In addition, a dichotomous variable of total anthocyanins was created to define high and low anthocyanin intake using the median as the cut-off point; median intake was equivalent to 19.14 mg/d.”

Please define mMDS in the methods.

Thank you, mMDS has been defined as the modified Mediterranean Diet Score in methods.

Modified text in methods: “Dietary intake was assessed at baseline using a validated 131-item semi-quantitative food-frequency questionnaire (FFQ), which reflected the previous year’s habitual intake, and a lifestyle questionnaire with additional dietary information, including a 13-item modified Mediterranean diet score (mMDS) (1,2).”

Was there a reason that women were included in the analyses despite the small sample size?

Lastly, with regard to our 5% female population, we have included a sensitivity analysis for multivariable adjusted linear regressions with the exclusion of females (Supplementary Table 4). Additionally, below we have provided a comparison of males and females, by excluding the opposite sex, in a less adjusted model that allowed conducting regressions on lipid measures with the 13 female participants.

Incorporated text in methods: “A sensitivity analysis was conducted with the additional exclusion of chronic diseases, women, and supplement use.”

Incorporated text in results: Sensitivity analysis for both exposures with additional exclusions for chronic disease, women, and supplement use further supported the robustness of our findings (Table S4).

Supplementary Table 4: Sensitivity analysis for anthocyanin intake (SD) and physical activity level on lipid profile measures (mg/dL).

Sensitivity Analysis

Anthocyanin intake (SD)

Physical activity level*

Exclusions

n

β (95%CI)

p-value

β (95%CI)

p-value

Chronic diseases†a

199

Triglycerides

-9.09 (-19.24 to 1.06)

0.08

-5.89 (-11.64 to -0.14)

0.05

Total cholesterol

-2.29 (-7.67 to 3.12)

0.41

-0.55 (-3.62 to 2.52)

0.72

HDL cholesterol

0.92 (-0.59 to 2.44)

0.23

0.85 (-0.01 to 1.70)

0.05

LDL cholesterol

-1.44 (-6.14 to 3.27)

0.55

-0.83 (-3.50 to 1.84)

0.54

LDL:HDL

-0.09 (-0.21 to 0.03)

0.16

-0.08 (-0.15 to -0.01)

0.03

TG:HDL

-0.27 (-0.58 to 0.04)

0.09

-0.21 (-0.39 to -0.04)

0.02

Total cholesterol:HDL

-0.14 (-0.29 to 0.01)

0.07

-0.10 (-0.18 to -0.01)

0.03

Womena

236

Triglycerides

-9.32 (-18.81 to 0.16)

0.05

-5.69 (-10.76 to -0.63)

0.03

Total cholesterol

-2.10 (-7.25 to 3.04)

0.42

-1.97 (4.72 to 0.77)

0.16

HDL cholesterol

0.72 (-0.67 to 2.12)

0.31

0.64 (-0.11 to 1.38)

0.09

LDL cholesterol

-1.01 (-5.58 to 3.55)

0.66

-1.97 (-4.40 to 0.46)

0.11

LDL:HDL

-0.08 (-0.20 to 0.03)

0.16

-0.09 (-0.15 to -0.03)

0.004

TG:HDL

-0.29 (-0.58 to 0.01)

0.06

-0.19 (-0.35 to -0.03)

0.02

Total cholesterol:HDL

-0.14 (-0.28 to 0.00)

0.05

-0.11 (-0.18 to -0.03)

0.004

Supplement use (proteins, glutamine, amino acids, etc.)a

170

Triglycerides

-8.72 (-19.39 to 1.96)

0.11

-4.79 (-10.38 to 0.80)

0.09

Total cholesterol

-4.22 (-10.53 to 2.08)

0.19

-3.56 (-6.84 to -0.29)

0.03

HDL cholesterol

0.38 (-1.36 to 2.11)

0.69

0.23 (-0.68 to 1.14)

0.61

LDL cholesterol

-2.77 (-8.44 to 2.91)

0.34

-2.83 (-5.78 to 0.12)

0.06

LDL:HDL

-0.09 (-0.22 to 0.05)

0.21

-0.07 (-0.14 to -0.00)

0.04

TG:HDL

-0.21 (-0.51 to 0.10)

0.18

-0.10 (-0.26 to 0.06)

0.21

Total cholesterol:HDL

-0.13 (-0.29 to 0.03)

0.12

-0.09 (-0.18 to -0.01)

0.03

*Physical activity was assessed using a scale of 0-7 representing levels of physical activity ranging from none to running >10 miles/wk or spending >3 hrs/wk in comparable physical activity.

Chronic diseases was defined as reporting a previous diagnosis or current treatment for hypertension, dyslipidemia, or diabetes, respectively.

aAdjusted for age, sex, BMI, total energy intake, mMDS, smoking status, education level, marital status, multivitamin use, supplement use, sleep, anthocyanin intake, prevalent hypertension, dyslipidemia, and type 2 diabetes

TG: triglycerides, HDL: high density lipoprotein cholesterol, LDL: low density lipoprotein cholesterol

Boldface indicates statistical significance (p<0.05)

Due to the statistical power among 13 female participants, the model here was the least adjusted multivariable model that adjusted for age, BMI, smoking status, education level, marital status, prevalent hypertension, dyslipidemia, and type 2 diabetes. We did not include this analysis in our findings due to the low statistical power for females and similar findings are presented in Supplementary Table 4 in a fully adjusted model with the additional exclusion of females.

Sensitivity Analysis

Anthocyanin intake (SD)

Physical activity level*

n

β (95%CI)

p-value

β (95%CI)

p-value

Mena

236

Triglycerides

-7.87 (-16.78 to 1.03)

0.08

-5.68 (-10.31 to -1.04)

0.02

Total cholesterol

-2.82 (-7.72 to 2.07)

0.26

-3.03 (-05.57 to -0.49)

0.02

HDL cholesterol

0.83 (-0.48 to 2.15)

0.22

0.62 (-0.06 to 1.31)

0.07

LDL cholesterol

-1.96 (-6.28 to 2.37)

0.37

-2.87 (-5.11 to -0.64)

0.01

LDL:HDL

-0.10 (-0.21 to 0.01)

0.07

-0.10 (-0.16 to -0.05)

<0.001

TG:HDL

-0.24 (-0.51 to 0.04)

0.09

-0.18 (-0.32 -0.04)

0.02

Total cholesterol:HDL

-0.15 (-0.28 to -0.02)

0.02

-0.12 (-0.01 to -0.06)

<0.001

Womena

13

Triglycerides

-6.15 (-27.75 to 15.45)

0.54

-11.57 (-36.74 to 13.59)

0.24

Total cholesterol

-34.78 (-54.18 to -15.39)

0.01

-12.85 (-37.91 to 12.21)

0.20

HDL cholesterol

0.42 (-22.2 to 23.04)

0.96

-0.19 (-11.85 to 11.47)

0.96

LDL cholesterol

-31.56 (-57.03 to -6.13)

0.03

-10.22 (-36.94 to 16.49)

0.31

LDL:HDL

-0.53 (-1.59 to 0.53)

0.21

-0.18 (-0.84 to 0.47)

0.44

TG:HDL

-0.29 (-1.79 to 1.21)

0.58

-0.23 (-0.93 to 0.46)

0.36

Total cholesterol:HDL

-0.59 (-1.87 to 0.69)

0.24

-0.23 (-0.98 to 0.52)

0.40

*Physical activity was assessed using a scale of 0-7 representing levels of physical activity ranging from none to running >10 miles/wk or spending >3 hrs/wk in comparable physical activity.

Chronic diseases was defined as reporting a previous diagnosis or current treatment for hypertension, dyslipidemia, or diabetes, respectively.

aAdjusted for age, BMI, smoking status, education level, marital status, prevalent hypertension, dyslipidemia, and type 2 diabetes.

TG: triglycerides, HDL: high density lipoprotein cholesterol, LDL: low density lipoprotein cholesterol

Boldface indicates statistical significance (p<0.05)

We have considered this characteristic of the firefighter working population with a very low percentage of females as a limitation of our study.

Modified text in discussion: Due to the predominately male prevalence of the firefighter profession, our results must be extrapolated to women with precaution. Moreover, this study population of Midwestern US career firefighters is not representative of the general population, however biological plausibility should be the basis for generalizations in epidemiology  (3,4).

Modified text in discussion: “Future research in this line conducted among a large representative population with the inclusion of additive interaction analysis could be instrumental for identifying narrower CVD risk subgroups and offer greater efficacy among behavior change interventions for the promotion of ideal cardiovascular health.”

REFERENCES:

  1. Willett WC, Sampson L, Stampfer MJ, Rosner B, Bain C, Witschi J, et al. Reproducibility and validity of a semiquantitative food frequency questionnaire. Am J Epidemiol [Internet]. 1985 [cited 2020 Jul 13];122(1):51–65. Available from: https://pubmed.ncbi.nlm.nih.gov/4014201/
  2. Yang J, Farioli A, Korre M, Kales SNSSN, Kales SNSSN. Modified Mediterranean Diet Score and Cardiovascular Risk in a North American Working Population. Gong Y, editor. PLoS One [Internet]. 2014 Feb 4 [cited 2017 Mar 12];9(2):e87539. Available from: https://dx.plos.org/10.1371/journal.pone.0087539
  3. Rothman K, Greenland S, Lash T. Modern Epidemiology. 3rd ed. Philadelphia, PA: Lippincott Williams and Wilkins; 2008.
  4. Rothman KJ, Gallacher JEJ, Hatch EE. Why representativeness should be avoided. Int J Epidemiol. 2013 Aug;42(4):1012–4.

Round 2

Reviewer 1 Report

I am satisfied with the responses from the authors. No more comments. Thanks for their efforts and careful revisions.